# The Rho-guanine nucleotide exchange factor PDZ-RhoGEF governs susceptibility to diet-induced obesity and type 2 diabetes

Ying-Ju Chang[1,2], Scott Pownall[1], Thomas E Jensen[3], Samar Mouaaz[1], Warren Foltz[4], Lily Zhou[1], Nicole Liadis[1], Minna Woo[5], Zhenyue Hao[1], Previn Dutt[1], Philip J Bilan[3], Amira Klip[3], Tak Mak[1†], Vuk Stambolic[1,2*†]

[1]Princess Margaret Cancer Center, University Health Network, Toronto, Canada; [2]Department of Medical Biophysics, University of Toronto, Toronto, Canada; [3]Cell Biology Program, The Hospital for Sick Children, Toronto, Canada; [4]Spatio-Temporal Targeting and Amplification of Radiation Response Program, Office of Research Trainees, University Health Network, Toronto, Canada; [5]Toronto General Research Institute, University Health Network, Toronto, Canada

*For correspondence: vuks@uhnresearch.ca

†These authors contributed equally to this work

Competing interests: The authors declare that no competing interests exist

**Abstract** Adipose tissue is crucial for the maintenance of energy and metabolic homeostasis and its deregulation can lead to obesity and type II diabetes (T2D). Using gene disruption in the mouse, we discovered a function for a RhoA-specific guanine nucleotide exchange factor PDZ-RhoGEF (*Arhgef11*) in white adipose tissue biology. While PDZ-RhoGEF was dispensable for a number of RhoA signaling-mediated processes in mouse embryonic fibroblasts, including stress fiber formation and cell migration, it's deletion led to a reduction in their proliferative potential. On a whole organism level, PDZ-RhoGEF deletion resulted in an acute increase in energy expenditure, selectively impaired early adipose tissue development and decreased adiposity in adults. PDZ-RhoGEF-deficient mice were protected from diet-induced obesity and T2D. Mechanistically, PDZ-RhoGEF enhanced insulin/IGF-1 signaling in adipose tissue by controlling ROCK-dependent phosphorylation of the insulin receptor substrate-1 (IRS-1). Our results demonstrate that PDZ-RhoGEF acts as a key determinant of mammalian metabolism and obesity-associated pathologies.

## Introduction

Obesity is a global health challenge. While it is generally recognized that obesity arises as a result of a complex interplay of genetic factors that govern predisposition for white adipose tissue expansion, as well as environmental factors, such as diet, the molecular mechanisms underlying this condition are not fully understood. Adipose tissue growth results from an increase in both adipocyte size and number and occurs as part of normal development, as well as during the onset of obesity (*Faust et al., 1978*; *Prins and O'Rahilly, 1997*). Recent evidence shows that the number of adipocytes is set during childhood and adolescence and determines the adult fat mass (*Spalding et al., 2008*). Adipogenesis, a differentiation program that gives rise to adipocytes, is regulated by a coordinated activation of a transcription factor network responding to the adipogenic and anti-adipogenic signals (*Cristancho and Lazar, 2011*). Insulin/IGF-1 signaling also has marked effects on adipogenesis. For instance, genetic disruption of the insulin receptor (IR), the insulin receptor substrate (IRS) genes, as well as PKBα/Akt1 and PKBβ/Akt2, respectively, the major mediators of insulin signaling, lead to impaired adipogenesis (*Miki et al., 2001*; *Tseng et al., 2004*; *Yun et al., 2008*).

**eLife digest** Obesity is a growing public health concern around the world, and can lead to the development of type 2 diabetes, heart disease and cancer. Both genetics and environmental factors such as diet contribute to obesity. Fat cells are essential to good health, but the excess accumulation of fat cells in obese people involves a complex process that is regulated by interactions between numerous genes, cellular messengers and mechanical forces. Learning more about these factors could help prevent or treat obesity.

One mutation in the gene encoding a protein called PDZ-RhoGEF has been linked to both obesity and type 2 diabetes. People with mutations in this gene are not responsive enough to insulin, a hormone important for sugar metabolism. This can interfere with the body's ability to burn energy in food or lead to a dangerous build up of sugar in the blood as seen in type 2 diabetes. But exactly what PDZ-RhoGEF normally does to prevent this is unclear.

Chang et al. now show that PDZ-RhoGEF controls fat cell production and the body's ability to release the energy contained in food. First, mice that had been genetically engineered to lack PDZ-RhoGEF were compared to typical mice. The mice without PDZ-RhoGEF had fewer fat cells than the typical mice, and they burned more energy. The mutant mice walked around about as much as the typical mice but they were more likely to have repetitive movements, the mouse equivalent of human nervous ticks.

Insulin normally stimulates the production of fat cells. But the mutant mice were less able to produce fat cells as they developed into adults. When fed a high fat food diet, the normal mice became fatter and insensitive to insulin and developed other health problems linked to excess fat in the body. The mutant mice on the same diet, however, stayed thin and avoided these health issues. The experiments show that PDZ-RhoGEF helps relay insulin's message within the body, and as such it plays a critical role in regulating metabolism, sugar levels and fat accumulation. Future work should ask how PDZ-RhoGEF affects other complications linked to obesity, and explore the possibility of developing treatments for obesity based on the biology of this molecule.

Beyond soluble factors, changes in cell shape have also been implicated in influencing adipogenesis through activation of a small GTPase RhoA and its downstream effector Rho-kinase (ROCK) (*Dupont et al., 2011*; *Kilian et al., 2010*; *McBeath et al., 2004*). RhoA-ROCK signaling is activated in mature adipocytes via mechanical stretch generated by obesity-related adipocyte hypertrophy (*Hara et al., 2011*). Together with Rac and Cdc42, RhoA constitutes a Rho branch of the Ras super-family of small GTPases, which act in a variety of signaling pathways and regulate many cellular functions (*Ridley, 2001a*). Initially identified as key regulators of actin cytoskeleton remodeling, Rho family members control many other cellular processes, including vesicular trafficking, cell cycle progression, cytokinesis, migration, apoptosis, gene transcription, and cell fate determination (*Coleman and Olson, 2002*; *Heasman and Ridley, 2008*; *McBeath et al., 2004*; *Sordella et al., 2002*).

Rho GTPases are activated through a GTP-for-GDP nucleotide exchange reaction, catalyzed by the Rho GTPase-specific guanine nucleotide exchange factors (RhoGEFs) (*Ridley, 2001a*). The Dbl family of RhoGEFs, with its 70 members in mammals, represents the largest class of Rho activators (*Rossman et al., 2005*; *Schmidt and Hall, 2002*). Dbl family members contain at least one highly conserved Dbl homology (DH) catalytic domain, which facilitates the GDP/GTP exchange, and a pleckstrin homology (PH) domain N-terminal to the DH domain (*Zheng, 2001*). PDZ-RhoGEF (ARH-GEF11), encodes a Rho guanine nucleotide exchange factor, with two transcript variants ARHGEF11 variant 1 and variant 2 in the human (GeneBank accession # NM_014784 and NM_198236) and a single Arhgef11 transcript in the mouse (Genebank accession NM_001003912). Together with two other Dbl family members, the leukemia-associated RhoGEF (LARG) and p115RhoGEF, PDZ-RhoGEF constitutes a newly identified subfamily, the regulator of G protein signaling (RGS) domain-containing RhoGEFs (RGS-RhoGEFs).

Single nucleotide polymorphisms (SNPs) within the genes encoding RGS-RhoGEFs have also been informative in revealing their potential physiological functions. For instance, a non-synonymous SNP (nsSNP) mapping to the C-terminal region of LARG and leading to Y1306C (Tyr1306Cys)

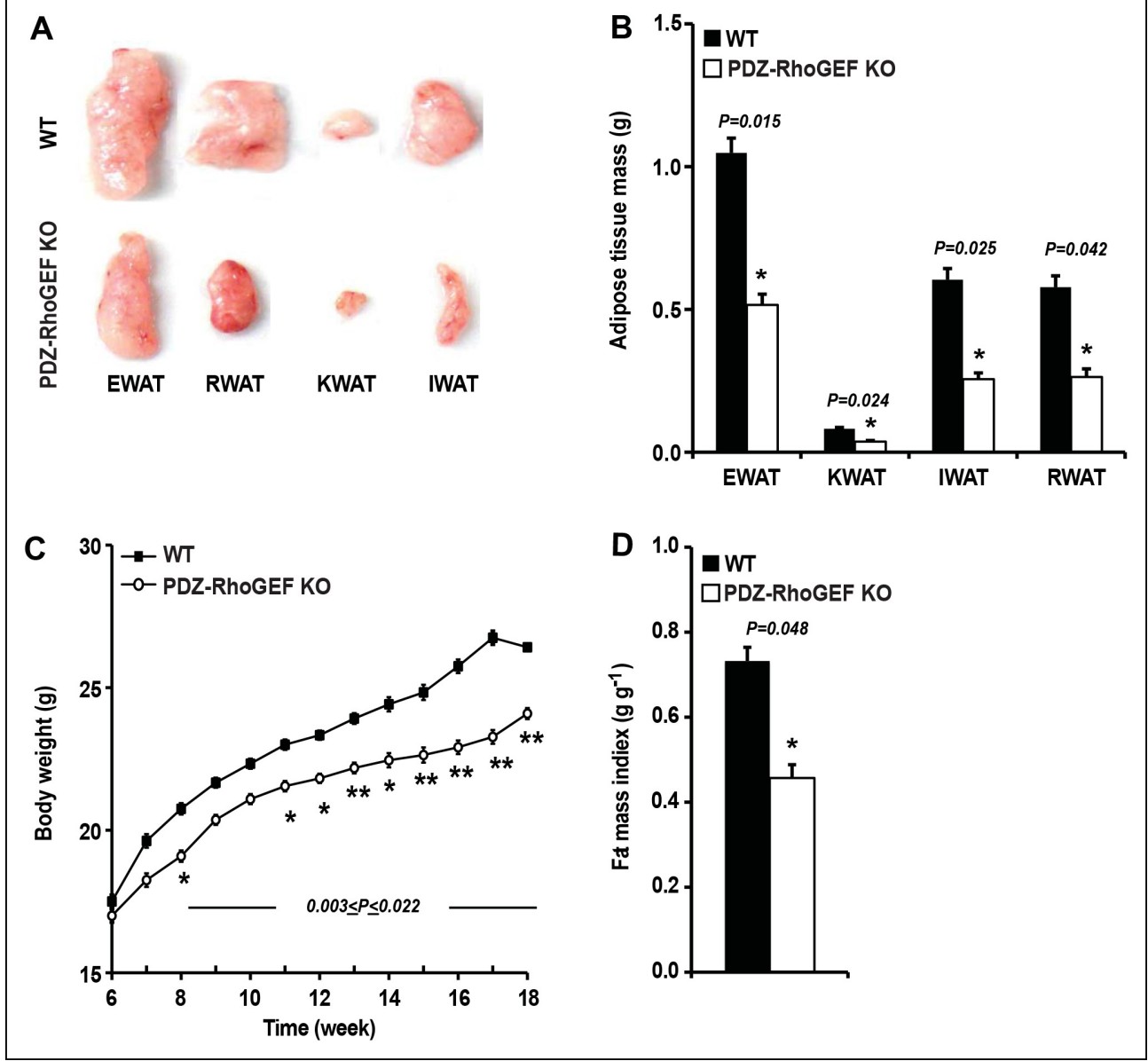

**Figure 1.** PDZ-RhoGEF controls body weight and adipose tissue mass. (**A**) Representative EWAT, RWAT, IWAT, and KWAT from NCD-fed 7-month old WT and PDZ-RhoGEF KO male mice. (**B**) Fat weight from 7-month old male mice on NCD (n = 7–10). (**C**)Body weight of mice on NCD (n = 12–13). (**D**) Body fat index determined by MRI analysis of wild type and PDZ-RhoGEF KO mice (n = 7–9) maintained on a NCD for 16 weeks. Statistical significance was calculated by a Student's t test using Excel software (*p≤0.05, **p≤0.01, ***p≤0.001).

The following figure supplements are available for Figure 1:

**Figure supplement 1.** PDZ-RhoGEF gene targeting in the mouse.

**Figure supplement 2.** Tissue mass of WT and PDZ-RhoGEF KO mice.

**Figure supplement 3.** The body weight, body size, and food intake of WT and PDZ-RhoGEF KO mice.

substitution has been implicated in increasing insulin sensitivity in non-diabetic Pima Indians (*Holzapfel et al., 2007*; *Kovacs et al., 2006*). Another nsSNP coding for the R1467H (Arg1467His), a C-terminal variant of PDZ-RhoGEF was found in association with insulin resistance and type II diabetes in several populations, including Pima Indians, Old Order Amish, Caucasian, Korean and

Chinese (*Bottcher et al., 2008*; *Fu et al., 2007*; *Jin et al., 2010*; *Liu et al., 2011*; *Ma et al., 2007*). These findings suggest that LARG and PDZ-RhoGEF may impact insulin sensitivity and whole body metabolism.

To elucidate the physiological function of PDZ-RhoGEF in vivo, we disrupted its gene in the mouse. While our data show that PDZ-RhoGEF is dispensable for many cellular processes that it had previously been linked to, we discovered its direct involvement in controlling adipose tissue homeostasis through regulation of adipocyte numbers and energy expenditure. Mechanistically, PDZ-RhoGEF was found to induce RhoA/ROCK-dependent phosphorylation of the insulin receptor substrate-1 (IRS-1) and promote signaling via this adaptor protein in response to insulin/IGF-1, controlling proliferation of adipogenic precursors and MEFs and whole body energy expenditure. Moreover, PDZ-RhoGEF-knockout (KO) mice were resistant to diet-induced obesity and insulin resistance in vivo. Our work firmly implicates PDZ-RhoGEF in the physiological control of adipose tissue development, obesity and insulin resistance.

## Results

### Reduced body weight and fat mass in mice lacking PDZ-RhoGEF

To evaluate the physiological function(s) of PDZ-RhoGEF, we generated a genetically modified mouse line bearing a conditionally targeted PDZ-RhoGEF allele, *Arhgef11* (*Figure 1—figure supplement 1A*). By introducing CMV-driven global expression of cre recombinase, PDZ-RhoGEF heterozygous mice (+/-) were generated and full deletion of PDZ-RhoGEF allele was confirmed by Southern Blot analysis (*Figure 1—figure supplement 1B*). We further verified the complete deletion of *Arhgef11* (-/-, KO) by immunoblotting with a rabbit polyclonal antibody raised against the N-terminal fragment of PDZ-RhoGEF (*Figure 1—figure supplement 1D*). PDZ-RhoGEF KO animals were born at the expected Mendelian ratio and were grossly indistinguishable from their wild type (WT) and heterozygous littermates.

Although PDZ-RhoGEF KO mice fully developed, as they aged, they weighed less than their wild type counterparts. Most of the tissues showed no difference in weight between the two genotypes (*Figure 1—figure supplement 2A*), however, the white adipose tissue mass was markedly reduced in mutant animals (*Figure 1A*). The fat pads from the major fat depots, epididymal (EWAT), knee (KWAT), inguinal (IWAT), and retroperitoneal white adipose tissue (RWAT) were significantly smaller in PDZ-RhoGEF KO compared to wild type mice, expressed both as total mass (*Figure 1B*) and as a percentage of body weight (*Figure 1—figure supplement 2B*). Unlike WAT, brown adipose tissue mass (BAT) was slightly affected by PDZ-RhoGEF deletion (*Figure 1—figure supplement 2C, D*).

To further assess the effect of PDZ-RhoGEF deletion on body weight, an age-matched set of 6 week-old wild type and PDZ-RhoGEF KO mice were monitored for weight gain for 12 weeks while on a normal chow diet (NCD) (5% of total calories derived from fat, 200 kcal/kg). Throughout the monitoring period, both male and female PDZ-RhoGEF KO mice weighed less then their wild type counterparts (*Figure 1C*, *Figure 1—figure supplement 3A*) while showing no difference in body length (*Figure 1—figure supplement 3B*) or food consumption (*Figure 1—figure supplement 3C*). Consistent with the anatomical analysis, MRI assessment of total fat of age matched 4-month old mice corrected for body weight, indicated that the body fat index for PDZ-RhoGEFKO mice was lower than that of the wild type (*Figure 1D*).

### The effect of PDZ-RhoGEF loss on energy expenditure

To investigate the changes underlying the weight reduction in PDZ-RhoGEF KO mice, we analyzed the brown adipose tissue (BAT), which plays an important role in the maintenance of energy homeostasis. The relative expression of the uncoupling protein-1 (UCP1) in BAT, the master regulator of BAT-mediated thermogenesis, was comparable between the genotypes (*Figure 2—figure supplement 1A*), as was the expression of UCP-1 in the subcutaneous WAT (IWAT) (a measure of the thermogenic UCP-1 positive brite/beige cells in WAT) (*Figure 2—figure supplement 1B*). Consistently, the animals from both genotypes showed no difference in core body temperature (*Figure 2—figure supplement 1C*).

We next examined the energy expenditure in our mice at different ages, over a 24 hr period, using indirect calorimetry, as well as the oxygen consumption rate (OCR) of isolated EWATs.

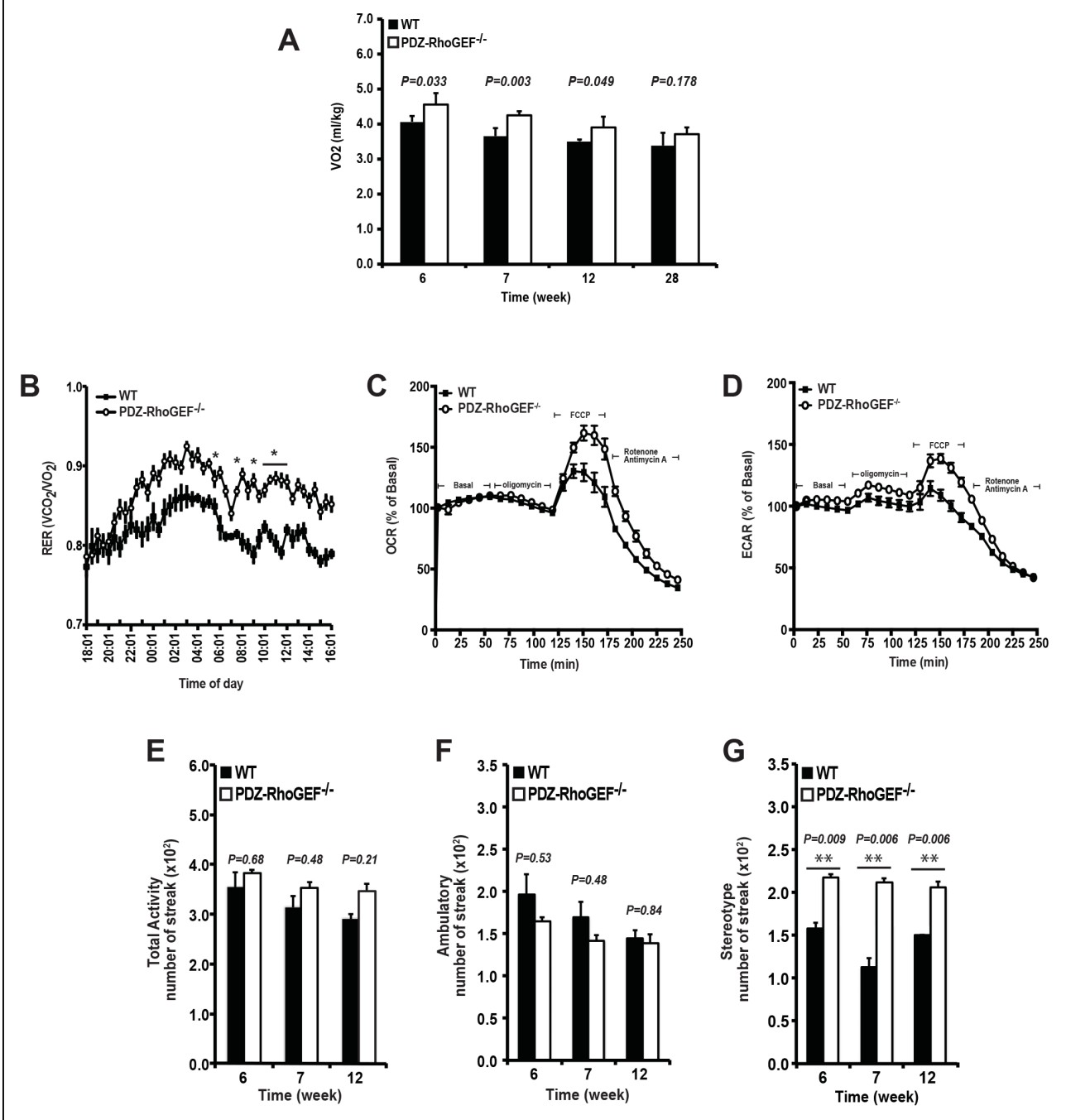

**Figure 2.** The effect of PDZ-RhoGEF on energy expenditure. (**A**) PDZ-RhoGEF KO mice consumed more oxygen than the wild type mice (n = 4). (**B**) PDZ-RhoGEF KO mice have higher RER than wild type mice (n = 4–6). (**C**) Oxygen consumption rate (OCR) of EWAT determined in the presence of mitochondrial effectors, oligomycin A (10 µM), FCCP (10 µM), and rotenone (3 µM)/antimycin A (12 µM). The values are presented as percent of the basal OCR measured at time 0 (n = 3). (**D**) Glycolysis was assessed by ECAR normalized to basal ECAR and measured at time 0 (n = 3). (**E–G**) Analysis of physical activity, including total activity (**E**), ambulatory (**F**), and stereotype activity (**G**). Statistical significance was calculated by a Student's t test using Excel software (*p≤0.05, **p≤0.01, ***p≤0.001).

The following figure supplements are available for Figure 2:

**Figure supplement 1.** UCP 1 expression in BAT and core body temperature.

**Figure supplement 2.** Energy homeostasis of male wild type and PDZ-RhoGEF KO mice at different ages.

Compared to wild type, the total oxygen consumption and the respiratory exchange rates (RER) in PDZ-RhoGEF KO mice were higher and particularly increased at time points towards the end of the light cycle (revised *Figure 2A,B*, *Figure 2—figure supplement 2A–C*). Consistently, while the isolated EWATs from both genotypes displayed similar basal OCR, PDZ-RhoGEF-deficient EWATs reached a higher maximal OCR than wildtype upon the injection of carbonyl cyanide-4-(trifluoromethoxy)-phenylhydrazone (FCCP), indicating their greater maximal respiratory capacity (*Figure 2C*). PDZ-RhoGEF KO EWATsalso displayed a higher extracellular acidification rate (ECAR), suggesting their preference for using glucose as a primary energy source (new *Figure 2D*).

Physical activity, a major determinant of total energy expenditure, was assessed by measuring total activity, including ambulatory and stereotype activity (grooming, digging, etc.). PDZ-RhoGEF KO animals displayed high total activity (*Figure 2E*). The locomotion, determined as ambulatory activity, was comparable between the genotypes (*Figure 2F*), whereas a trend towards increased stereotype activity in PDZ-RhoGEF KO animals was detected (*Figure 2G*), contributing to an upward trend in total activity (*Figure 2E*).

## Reduction of adipocyte and adipocyte progenitor abundance in PDZ-RhoGEF KO EWATs

Considering that adipocyte number and adipocyte cell size can impact fat mass, we examined the adipose tissue properties of our mice. Morphometric analysis of fixed adipose tissue from the EWAT of wild type and PDZ-RhoGEF KO animals showed no discernable differences in adipocyte size between the genotypes (*Figure 3—figure supplement 1A*). However, the number of mature adipocytes isolated from the EWAT of PDZ-RhoGEF KO mice was lower compared to that from wild type animals at various ages (*Figure 3A*). Animals from both genotypes displayed comparable adipogenic potential, judged by the expression levels of the peroxisome proliferator-activated receptor γ (PPARγ), a key determinant of adipocyte terminal differentiation, and adiponectin (Acrp30) in EWAT (*Figure 3—figure supplement 1B*). Moreover, PDZ-RhoGEF-deficient adipose tissue stromal cells (ADSCs) and MEFs retained the ability to differentiate into mature adipocytes, albeit at a lower rate (*Figure 3B*, *Figure 3—figure supplement 1C*).

PDZ-RhoGEF KO MEFs displayed impaired cell proliferation, both under normal growth conditions (*Figure 3—figure supplement 2A*) and in response to fetal calf serum (FCS) and Insulin-like Growth Factor-1 (IGF-1) (*Figure 3—figure supplement 2B*), as did the PDZ-RhoGEF KO ADSCs in response insulin (*Figure 3C*). Similarly, lowered expression of PDZ-RhoGEF in a preadipocyte cell line 3T3-L1 reduced their proliferative capacity under normal culture conditions, as well as in response to insulin (*Figure 3—figure supplement 2C,D*)

IGF-1/insulin signaling has been implicated in the control of the last proliferative phase of adipogenesis, also known as mitotic clonal expansion (MCE) (*Avram et al., 2007*). To further address the effect of PDZ-RhoGEF on MCE, post-confluent mitosis and clonal expansion were monitored by [³H]-thymidine uptake and pulse-chase bromodeoxyuridine (BrdU) labeling in MEFs after induction of adipogenesis. PDZ-RhoGEF KO MEFs displayed greatly diminished DNA synthesis and cell proliferation compared to wild type (*Figure 3—figure supplement 3A*). Finally, ectopic expression of PDZ-RhoGEF promoted DNA synthesis during clonal expansion of 3T3-L1 cells (*Figure 3—figure supplement 3B*). Taken together, these data implicate PDZ-RhoGEF in coupling insulin/IGF-1 signaling to the proliferative phases of adipose tissue development.

The adipogenic progenitor cells exist within the stromal vasculature fraction (SVF) of WAT (*Rodeheffer et al., 2008*) and are represented by the Lin⁻/CD29⁺/CD34⁺/Sca1⁺/CD24⁺ (CD24⁺) and the Lin⁻/CD29⁺/CD34⁺/Sca1⁺/CD24⁻ (CD24⁻) populations, which we isolated from our mice (*Figure 3—figure supplement 4*). Judged by BrdU incorporation, deletion of PDZ-RhoGEF led to a reduction of proliferative capacity in cultured adipogenic progenitors (*Figure 3D*). To probe the proliferative capacity of the adipose tissue in vivo during neonatal development, wild type and PDZ-RhoGEF KO postnatal day 6.5 mice were injected with BrdU at 12 hr intervals for 3.5 days. At one-month of age, mice were sacrificed and the BrdU-retaining cells in the adipose tissue counted. PDZ-RhoGEF KO EWAT displayed fewer BrdU-retainingcells than the wild type EWAT, indicative of impaired proliferation (*Figure 3E*). Of note, EWAT from both genotypes displayed the same proportion of infiltrating macrophages, determined by staining for the macrophage marker F4/80 (*Figure 3—figure supplement 5A*), eliminating the possibility that the differences in the BrdU signal arose from variable infiltrates. To pursue the possible role of PDZ-RhoGEF in regulation of adipose

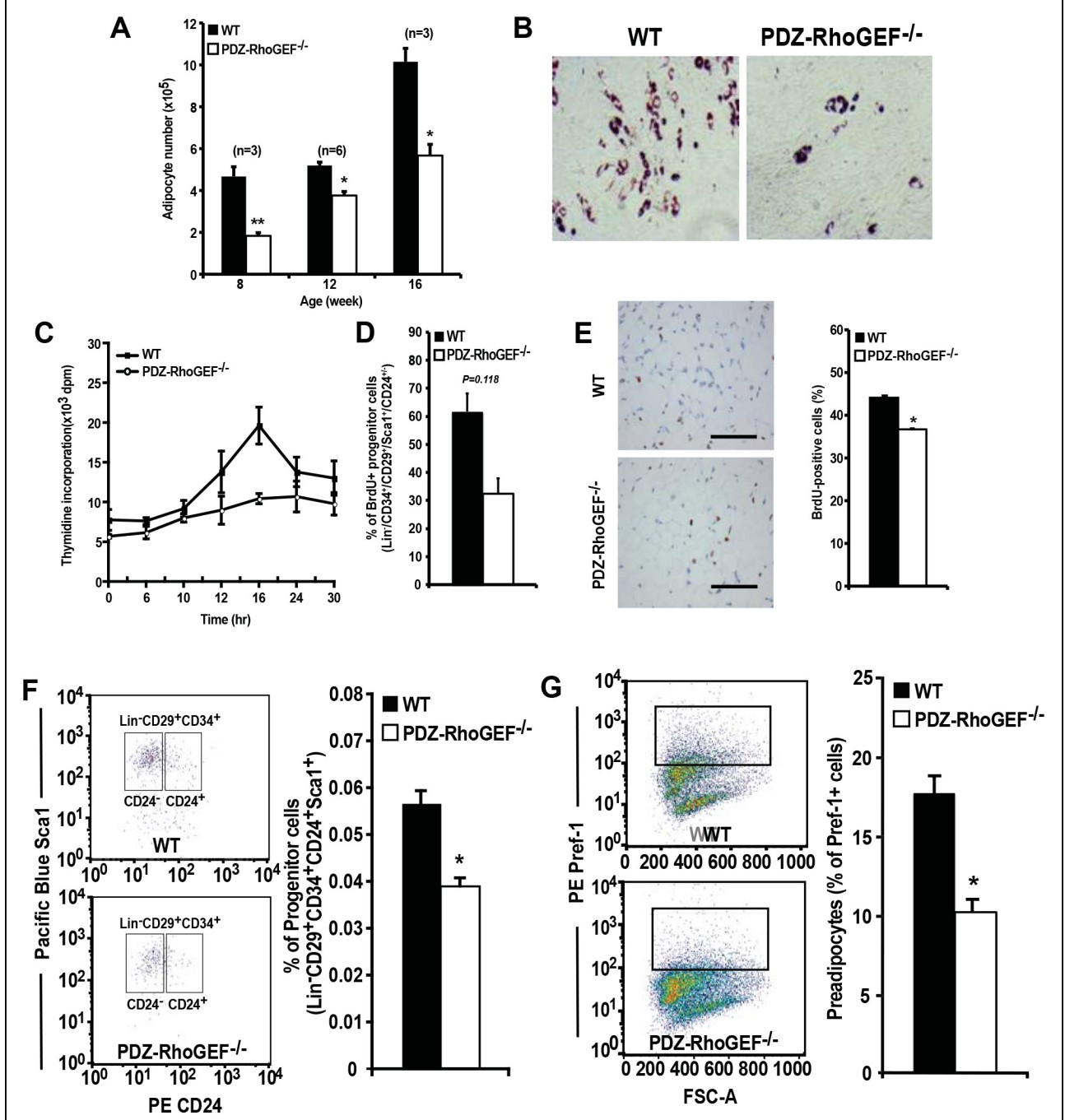

**Figure 3.** Reduced adipocytes and progenitors in PDZ-RhoGEF KO mice. (**A**) Adipocyte numbers from 8-, 12-, 16-week old WT and PDZ-RhoGEF KO EWATs, (**B**) In vitro adipogenesis of EWAT ADSCs prepared from 16 week-old WT and PDZ-RhoGEF KO animals assessed by Oil Red-O staining. (**C**) DNA synthesis of EWAT-derived ADSCs upon insulin stimulation during MCE. (**D**) Adipogenitc progenitor cell proliferation presented as percent of BrdU + cells by FACS analysis under normal culture condition (n = 3). (**E**) Image and ImageJ quantification of BrdU-labeled EWATs from one-month old mice (n = 6–9). Bar scale = 100 µm (200X). Statistical significance was calculated by a Student's t test using Excel software (*$p \leq 0.05$, **$p \leq 0.01$, ***$p \leq 0.001$). (**F**) Stromal-vascular fraction from adult WT and PDZ-RhoGEF KO EWATs. EWAT was stained with adipogenic progenitor cells (CD29+/CD34+/CD24+/Sca1+) (n = 4). (**G**) Preadipocytes in WT and PDZ-RhoGEF KO EWAT stromal-vascular fraction were stained with preadipocyte marker, pref-1 (n = 3). Statistical significance was calculated by a Student's t test using Excel software (*$p \leq 0.05$, **$p \leq 0.01$, ***$p \leq 0.001$).

The following figure supplements are available for Figure 3:

**Figure supplement 1.** Cell size, adipogenic potential, and in vitro adipogenesis.

*Figure 3. continued on next page*

precursor population, we quantified the adipocyte progenitor cell (Lin⁻/CD29⁺/CD34⁺/Sca1⁺/CD24⁺) and preadipocyte (Pref-1⁺) populations within the SVF of the adult mouse fat pads. Both the adipogenic progenitor and preadipocyte populations were reduced in PDZ-RhoGEF KO animals (*Figure 3F,G*), accompanied by a proportional decrease in mRNA expression of Pref-1 (*Figure 3—figure supplement 5B*).

## PDZ-RhoGEF modulates PI3K signaling via regulation of ROCKs

We explored a possible general function of PDZ-RhoGEF in controlling proliferative capacity. The defective proliferation of PDZ-RhoGEF-deficient MEFs (*Figure 3—figure supplement 3A,B*) was paralleled by reduced Rho activation upon FCS and IGF-1 treatment (*Figure 4A*). Of interest, the LPA-dependent Rho stimulation was comparable between the genotypes, reinforcing that PDZ-RhoGEF is not required for LPA-mediated RhoA activation (*Figure 4A*). Moreover, RhoA-mediated actin cytoskeleton reorganization, stress fiber formation and migration upon exposure to LPA, FCS, and IGF-1 were unaffected by PDZ-RhoGEFloss (*Figure 4—figure supplement 1A–C*).

To probe the function of PDZ-RhoGEF in mediating IGF-1 sensitivity, we tested the effect of PDZ-RhoGEF loss on PKB/Akt signaling. PDZ-RhoGEF KO MEFs, as well as the cultured FACS-sorted adipogenic progenitor cells, exhibited markedly lower PKB/Akt activation in response to IGF-1 (*Figure 4B,C*). Re-expression of PDZ-RhoGEF in the PDZ-RhoGEF-deficient MEFs rescued the IGF-1-induced PKB/Akt phosphorylation (*Figure 4D*). However, activation-specific tyrosine phosphorylation of the IGF-1 receptor was comparable between the two genotypes (*Figure 4E*), indicating that diminution of PKB/Akt phosphorylation in PDZ-RhoGEF KO MEFs stems from an impaired PDZ-RhoGEF-dependent step downstream of the receptor. Insulin receptor substrate-1 (IRS-1) acts immediately downstream of the insulin/IGF-1 receptors to propagate their signals (*Taniguchi et al., 2006*). Furthermore, IRS-1 S632/635 have been implicated as substrates of ROCKI/II (*Furukawa et al., 2005*; *Lee et al., 2009*). In IGF-1-stimulated PDZ-RhoGEF KO MEFs, both tyrosine and IRS1 serine 632/635 (S632/635) phosphorylation were greatly reduced compared to wild type MEFs, as was its direct association with the p85α and p110α subunits of PI3K (*Figure 4F*). In contrast, phosphorylation of IRS-1 at S612, another putative Rho kinase (ROCKI/II) targeted residue (*Sordella et al., 2003*), was not affected by PDZ-RhoGEF loss (*Figure 4F*), indicating that phosphorylation of this residue in MEFs can be uncoupled from that of S632/635. When wild type MEFs were pretreated with Y27632, a specific inhibitor of ROCKI and ROCKII, IGF-1-dependent PKB/Akt phosphorylation was reduced to a level comparable to that found in PDZ-RhoGEF KO MEFs (*Figure 4G*), suggesting that these kinases mediate the PDZ-RhoGEF-dependent IGF-1 response. Thus, PDZ-RhoGEF modulates IGF-1-mediated cell proliferation in MEFs via regulation of ROCK-dependent phosphorylation of IRS-1 at S632/635.

As one of the peripheral insulin targeted tissues, fat mass is associated with insulin sensitivity and metabolic homeostasis (*Guilherme et al., 2008*). Upon IP injection of insulin, activation-specific tyrosine phosphorylation of the insulin receptor (IR) was comparable between wild type and PDZ-RhoGEF KO EWAT (*Figure 5A*). However, activation of PKB/Akt in EWAT from PDZ-RhoGEF KO animals was considerably reduced (*Figure 5A*), consistent with a role for PDZ-RhoGEF in the transmission of the insulin signal downstream of IR and reminiscent of the MEFs response to IGF-1 (*Figure 4B,E*). Further, both tyrosine and S632/635 phosphorylation of IRS-1 in PDZ-RhoGEF KO EWAT were decreased in response to insulin (*Figure 5B*), as was the direct association of IRS-1 with the p85α and p110α subunits of PI3K (*Figure 5B*), indicative of impeded insulin signaling throughput via IRS-1 (*Furukawa et al., 2005*; *Lee et al., 2009*). In accordance with the MEF data, phosphorylation of S612 of IRS1 remained similar between the genotypes (*Figure 5B*). Of note, PDZ-RhoGEF loss did not affect insulin-induced PKB/Akt phosphorylation in two other peripheral insulin target tissues, the liver (*Figure 5C*), where PDZ-RhoGEF is abundant (*Figure 5—figure supplement 1*), or the skeletal muscle (*Figure 5D*), where PDZ-RhoGEF expression is relatively low (*Figure 5—figure supplement 1*).

## PDZ-RhoGEF contributes to HFD-induced obesity and type II diabetes (T2D)

Despite the reduction of fat mass, insulin-mediated glucose excursion was not affected in PDZ-RhoGEF KO mice, as determined by the glucose tolerance test (GTT) (*Figure 6—figure supplement 1A*), nor was the insulin-dependent glucose transport in the soleus and EDL (extensor digitorum

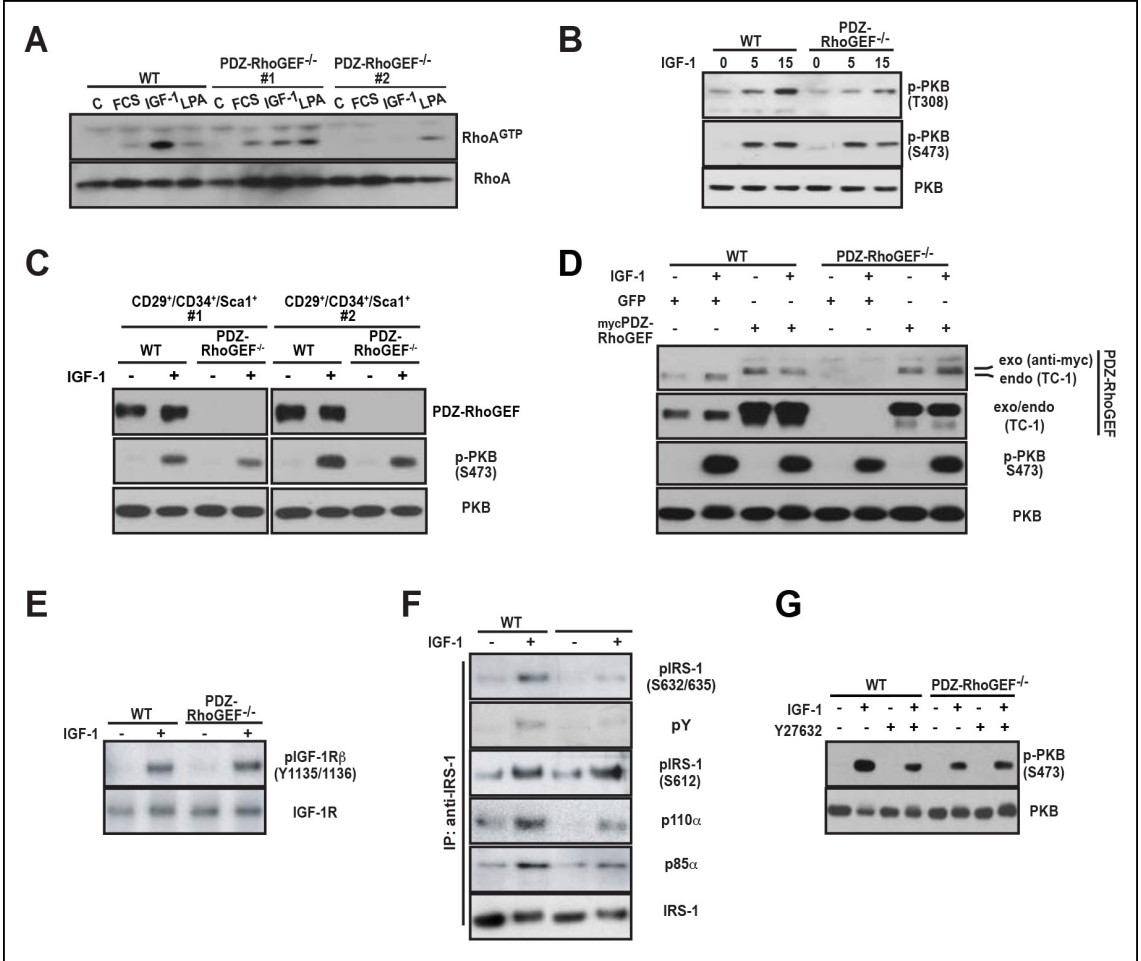

**Figure 4.** PDZ-RhoGEF is required for the optimal IGF-1 signaling output in MEFs and adipogenic progenitor cells. (**A**) RhoA activation was determined by the RBD-binding assay in response to the indicated factors (FCS [10%], IGF-1 [100 ng/ml], LPA [20 μM]). (**B**) Differential PKB/Akt phosphorylation in WT and PDZ-RhoGEF KO MEFs in response to IGF-1. (**C**) Differential PKB/Akt phosphorylation and PDZ-RhoGEF expression in WT and PDZ-RhoGEF KO adipogenic progenitor cells in response to IGF-1. (**D**) IGF-1 response of PDZ-RhoGEF KO MEFs is rescued by ectopic expression of myc-tagged PDZ-RhoGEF. (**E**) IGF-1 stimulated tyrosine phosphorylation of IR. (**F**) Serine phosphorylation of IRS1 and interaction between IRS1 and PI3K in WT and PDZ-RhoGEF KO MEFs in response to IGF-1. (**G**) Phosphorylation of PKB/Akt S473 in response to IGF-1 is dependent on ROCK activity.

The following figure supplements are available for Figure 4:

**Figure supplement 1.** Stress fiber formation and mitogen-induced cell migration.

longus) skeletal muscles (***Figure 6—figure supplement 1B***). To further elucidate the effect of PDZ-RhoGEF on glucose utilization in the liver, we monitored the glucose-dependent gene expression of the carbohydrate response binding protein (ChREBP) and the glucose transporter 2 (Glut2), and found no difference between the wild type and PDZ-RhoGEF KO livers (***Figure 6—figure supplement 1C***). Furthermore, there was no lipid accumulation in the liver (***Figure 6—figure supplement 1D***).

To probe if the effect of PDZ-RhoGEF on adipocyte numbers and energy expenditure can impact the onset of diet-induced obesity and T2D, two age groups of mice (6-week old and 16-week old) were transferred to a high fat diet (HFD) (***Supplementary file 1***) for 14 to 16 weeks. Wild type and PDZ-RhoGEF KO mice from both age groups gained weight at similar rates upon switching to HFD (***Figure 6A***, ***Figure 6—figure supplement 2A***). Nevertheless, the wild type mice from both age groups retained a proportionally higher body weight and higher adiposity (***Figure 6—figure supplement 2B, C***), despite consuming equal amounts of food (***Figure 6—figure supplement 2D***). Notably, the animals from both genotypes showed no difference in total oxygen consumption, RER or

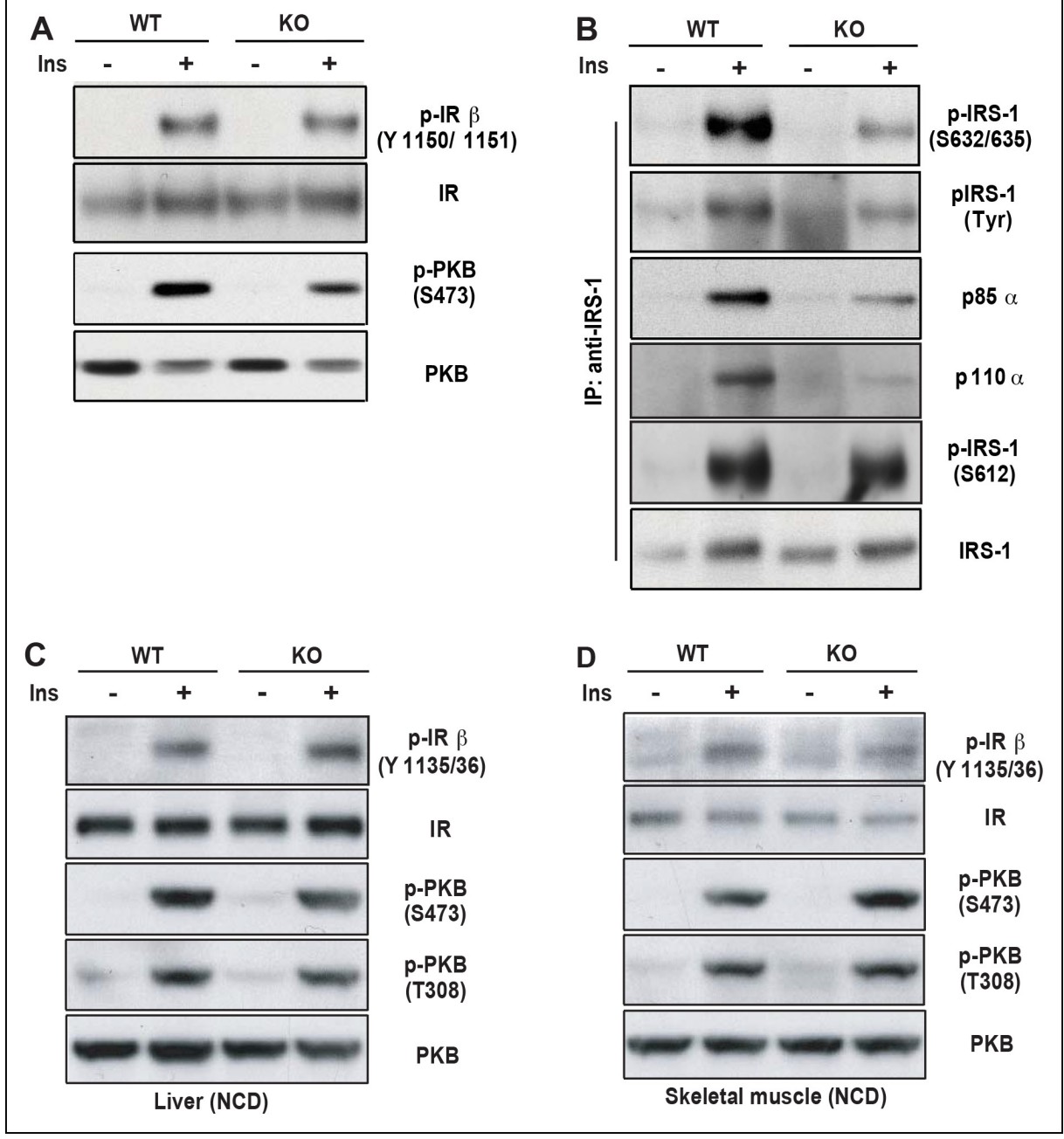

**Figure 5.** PDZ-RhoGEF is required for optimal IGF-1 signaling output in EWAT. (**A**) IR phosphorylation and differential PKB/Akt phosphorylation in WT and PDZ-RhoGEF KO EWAT in response to insulin. (**B**) Serine phosphorylation of IRS1 and interaction between IRS1 and PI3K in WT and PDZ-RhoGEF KO EWAT in response to insulin. (**C**) IR phosphorylation and differential PKB/Akt phosphorylation in WT and PDZ-RhoGEF KO liver in response to insulin. (**D**) IR phosphorylation and differential PKB/Akt phosphorylation in WT and PDZ-RhoGEF KO skeletal muscle in response to insulin.

The following figure supplements are available for Figure 5:

**Figure supplement 1.** The tissue expression profile of PDZ-RhoGEF.

total locomotor activity (*Figure 6—figure supplement 3A–C*). Importantly, wild type but not PDZ-RhoGEF KO mice, developed glucose intolerance (*Figure 6B*, *Figure 6—figure supplement 4A*) and insulin resistance (*Figure 6C*, *Figure 6—figure supplement 4B*), as well as impaired insulin-dependent glucose transport in the soleus and EDL (*Figure 6D*). Further, wild type EWAT exhibited higher frequency of larger adipocytes (*Figure 6E*), indicative of adipocyte hypertrophy, which is

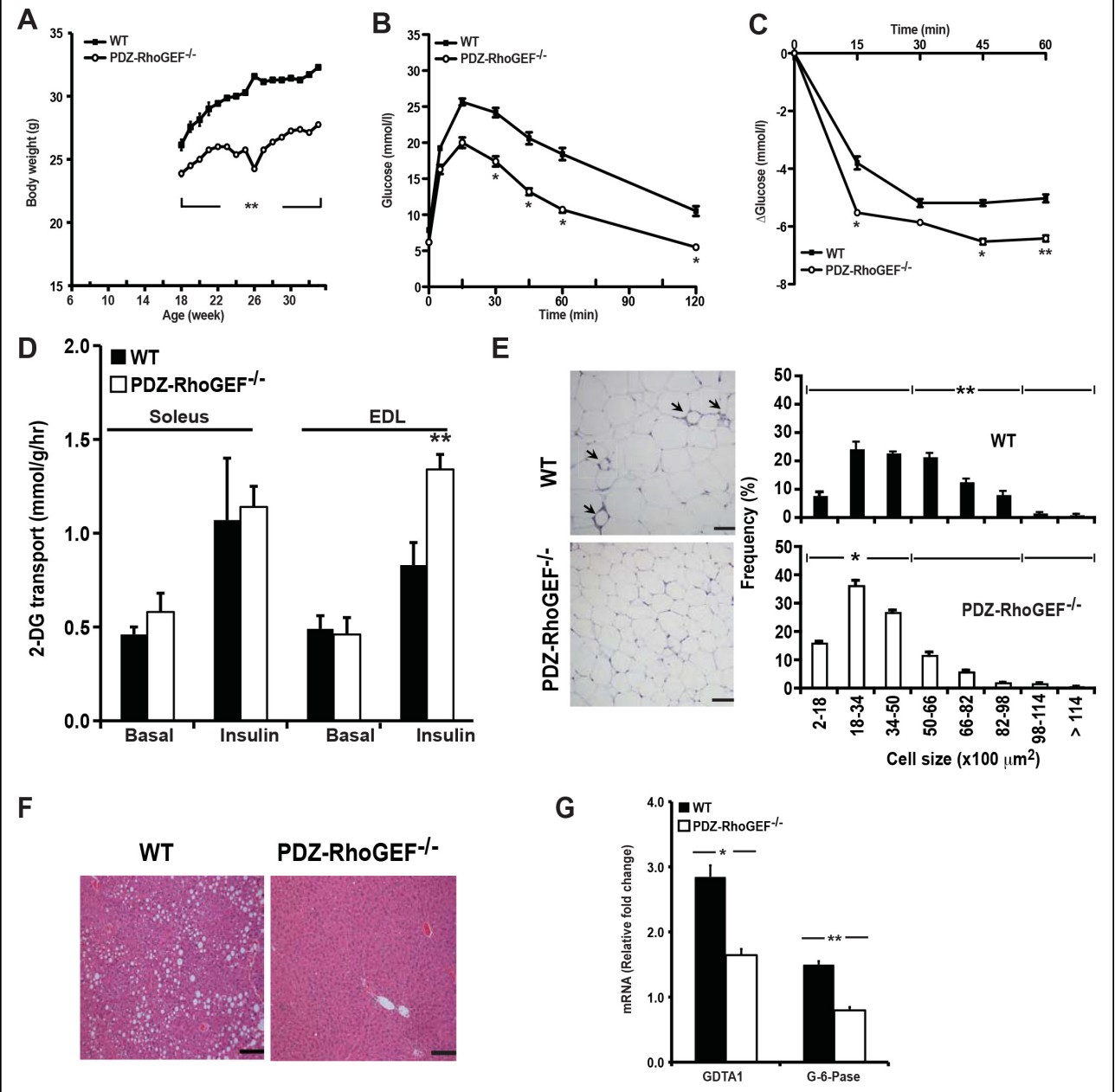

**Figure 6.** PDZ-RhoGEF deficiency protects animals from diet-induced obesity, glucose intolerance and insulin resistance. (**A**) The body weight of mice switched to HFD at 18 weeks of age was monitored for 14 weeks (n = 8). (**B**) GTT on overnight fasted 32-wk old mice after 14 weeks on HFD (n = 4). (**C**) Insulin-mediated glucose excursion in mice described in (**B**) determined by ITT (n = 8). (**D**) Ex vivo glucose transport in isolated skeletal muscles (Soleus and EDL) from HFD-fed animals (n = 4). Statistical significance was calculated by a Student's t test using Excel software (*p≤0.05, **p≤0.01, ***p≤0.001). (**E**) Morphometric analysis of epididymal adipocyte cell size and crown-like structure (arrows) in WT and PDZ-RhoGEF KO mice (n = 4–6). Scale bar = 100 μm (100X). (**F**) Histological analysis of representative liver sections from WT and PDZ-RhoGEF KO mice. Scale bar = 100 μm (100X). (**G**) The mRNA expression of DGAT1 and G-6-Pase in liver from HFD-fed animals (n = 4). Statistical significance was calculated by a Student's t test using Excel software (*p≤0.05, **p≤0.01, ***p≤0.001).

The following figure supplements are available for Figure 6:

**Figure supplement 1.** Glucose metabolism in WT and PDZ-RhoGEF KO mice under normal diet.

**Figure supplement 2.** Adiposity and food consumption in WT and PDZ-RhoGEF KO mice under high-fat diet.

*Figure 6. continued on next page*

associated with a decrease in adiponectin secretion (*Table 1*) and increased local inflammation with macrophage infiltration, detected as crown-like structures (CLS) (*Figure 6E*).

While PDZ-RhoGEF KO livers remained normal, wild type mice developed non-alcoholic fatty liver (NAFL) (*Figure 6F*), a pathological consequence of ectopic fat deposition due to adipose hypertrophy. In wild type, but not PDZ-RhoGEF KO mice on HFD, this was accompanied by increased mRNA expression of diglyceride acyltransferase 1 (DGAT1), an enzyme that participates in triglyceride synthesis (*Figure 6G*), Unlike PDZ-RhoGEF KO animals, wt mice displayed elevated mRNA levels of glucose-6-phosphatase (G-6-Pase), a key enzymes involved in at final step of gluconeogenesis and glycogenolysis in liver (*Figure 6G*), suggesting that increased glucose level in wild type mice may be partly due to elevated hepatic glucose production (*Table 1*) (*Elam et al., 2010*; *Higuchi et al., 2008*; *Pettinelli et al., 2009*; *Villanueva et al., 2009*). Eventually, wild type mice developed T2D with elevated fasting glucose and insulin (*Table 1*), while the PDZ-RhoGEF KO animals maintained normal glucose homeostasis (*Figure 6—figure supplement 4C*).

## PDZ-RhoGEF affects insulin resistance through a S6K1-mediated negative feedback loop

We further investigated the molecular mechanism of how PDZ-RhoGEF contributes to HFD-induced insulin resistance and T2D. In accordance with the impaired whole body insulin resistance, peripheral insulin-target tissues from wild type mice displayed diminished PKB/Akt phosphorylation on HFD (*Figure 7A*). Nevertheless, total IR levels and its tyrosine phosphorylation remained similar in livers and EWAT from both genotypes on HFD (*Figure 7—figure supplement 1*), suggesting that the insulin resistance in these tissues is not due to IR desensitization or degradation.

We next explored inhibitory serine phosphorylation of IRS1 as the source of insulin resistance downstream of IR in the white adipose tissue (*Griffin et al., 1999*; *Hotamisligil et al., 1996*). Under HFD, phosphorylation of IRS1 S632/635 and S307, targets of c-Jun N-terminus kinase 1 (JNK1) and mTOR/S6K1, were elevated in wild type EWAT (*Figure 7B*). While JNK1 activation appeared unaffected by PDZ-RhoGEF deletion, p70S6K1 activation was considerably higher in wild type EWAT under HFD (*Figure 7B*). Even on normal food, insulin-mediated p70S6K1 activation was reduced in PDZ-RhoGEF KO EWAT (*Figure 7C*). Intriguingly, prolonged HFD not only aggravated the activation of p70S6K signaling but also led to an elevation of PDZ-RhoGEF protein levels in all peripheral insulin-targeted tissues, EWAT, skeletal muscle (EDL), and liver (*Figure 7D*), suggesting that high levels of PDZ-RhoGEF accompany HFD-induced insulin resistance in wild type animals.

## Discussion

The 20 members of the Rho family of GTPases are regulated by more than 60 GEFs and 70 GAPs, and are thought to signal downstream through more than 60 effectors (*Bishop and Hall, 2000*; *Schmidt and Hall, 2002*; *Tcherkezian and Lamarche-Vane, 2007*). While such complexity leads to functional redundancy in the case of certain signals, various regulatory inputs display differential specificity toward particular Rho family members and act in a context-dependent manner (*Etienne-Manneville and Hall, 2002*; *Ridley, 2001a*). PDZ-RhoGEF was initially identified as an activator of RhoA acting downstream of GPCRs through association with $G\alpha12$ or $G\alpha13$ (*Fukuhara et al., 1999*), and subsequently found as a component of other signaling pathways implicated in cell migration,

**Table 1.** Quantification of fasting serum metabolites of wild type and PDZ-RhoGEF KO mice.

|  | Wild type | PDZ-RhoGEF KO |
|---|---|---|
| Triglycerides (mg/ml) | 0.63 +±0.02 | 0.43 + ± 0.01* |
| Glucose (mmol/l) | 12.13 + ± 0.53 | 7.62 + ± 0.16* |
| Insulin (ng/ml) | 1.80 + ± 0.20 | 0.51 + ± 0.07* |
| Adiponectin (ng/ml) | 0.29 + ± 0.01 | 0.36 + ± 0.01* |

Values are from 8-month-old male mice of the indicated genotype fed with high fate diet after overnight fasting.

Data represent mean + ± s.e.m.

*$p<0.05$ compared with wild type (n = 8).

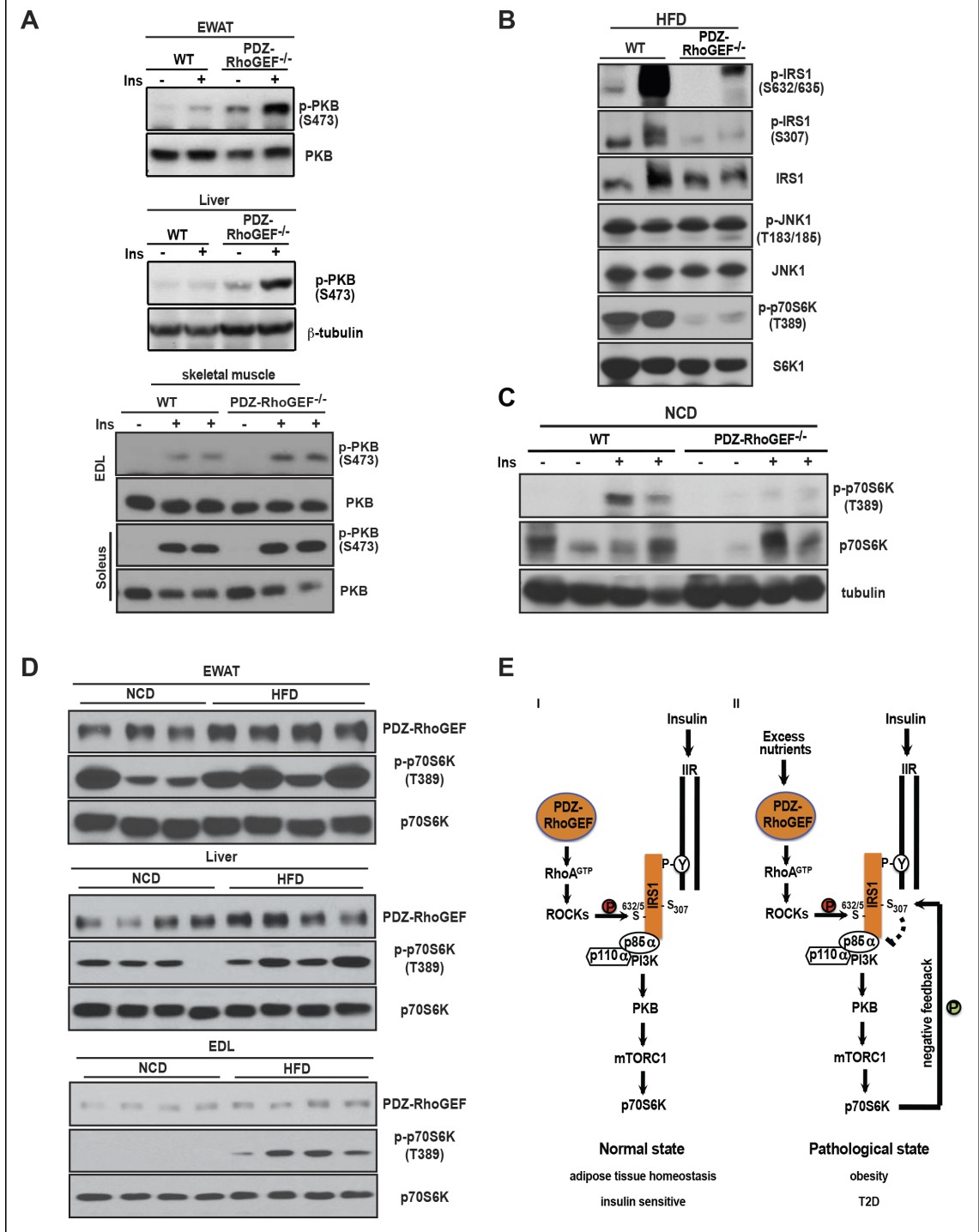

**Figure 7.** PDZ-RhoGEF contributes to the HFD-induced insulin resistance through a S6K1-mediated negative feedback. (**A**) Western blot analysis of insulin-mediated PKB/Akt phosphorylation in peripheral tissues: EWAT, liver, and skeletal muscles (EDL and soleus). (**B**) Phosphorylation status of IRS1 S632/635, S307, JNK1 T183/185, and p70S6K1 T389 in EWATs from HFD-fed WT and PDZ-RhoGEF KO mice. (**C**) Phosphorylation of p70S6K1 T389 in EWATs from NCD-fed WT and PDZ-RhoGEF KO mice after in vivo insulin stimulation. (**D**) Differential protein levels of PDZ-RhoGEF in EWAT of NCD- and HFD-fed wild type mice. (**E**) PDZ-RhoGEF modulates insulin/IGF-1 signaling to impact adipose tissue homeostasis (**I**) and susceptibility to dietary-induced obesity and T2D (**II**).

The following figure supplements are available for Figure 7:

**Figure supplement 1.** EWAT insulin sensitivity determined by IR tyrosine phosphorylation.

axon outgrowth, and chemotaxis (*Driessens et al., 2002*; *Kuner et al., 2002*; *Perrot et al., 2002*; *Zheng et al., 2006*). Accumulating evidence indicates that PDZ-RhoGEF, LARG and p115RhoGEF transmit a partially overlapping set of signals to RhoA (*Chikumi et al., 2004*; *Mikelis et al., 2013*). With its restricted expression in the hematopoietic system, p115RhoGEF was found to be required for marginal zone B cell and leukocyte homeostasis (*Francis et al., 2006*; *Girkontaite et al., 2001*; *Wirth et al., 2008*), whereas LARG plays a major role in maintaining the $G_{12}$-$G_{13}$–mediated vascular smooth muscle tone during the development of salt-dependent hypertension, consistent with its predominant expression in the aorta media (*Wirth et al., 2008*). Furthermore, PDZ-RhoGEF, p115RhoGEF and LARG also display overlapping roles in linking $G_{\alpha12/13}$-coupled GPCRs to RhoA activation, cell migration, and thrombin-mediated cell proliferation (*Mikelis et al., 2013*). Using genetic disruption in the mouse, we have shown that PDZ-RhoGEF is dispensable for embryo development and a number of known RhoA signaling-mediated processes, including stress fiber formation and cell migration in fibroblasts (*Ridley, 2001b*). Nevertheless, PDZ-RhoGEF has a prominent function in modulating insulin signaling in mouse adipose tissue, with profound effects on whole body physiology, including development of obesity and obesity-associated T2D.

## PDZ-RhoGEF regulates IRS-1/PI3-kinase signaling

In search of a molecular mechanism for PDZ-RhoGEF action, we discovered differential signaling at the level of IRS-1 as the likely cause for the suboptimal proliferative signaling in the absence of PDZ-RhoGEF. Differential S632/635 IRS-1 phosphorylation and association with PI3K, as well as PKB/Akt activation in MEFs (*Figure 4B,E*) and EWAT (*Figure 5A,B*), points to a mechanism of PDZ-RhoGEF/RhoA impact on insulin/IGF-1 signaling involving ROCKs. Indeed, inhibition of ROCKs in wild type MEFs impaired their IGF-1 response (*Figure 4F*).

With 182 serine and 60 threonine residues, IRS-1 has been shown to be a target of many kinases, including ROCKs, JNK (c-Jun NH2-terminal kinase), mTOR (mammalian target of rapamycin), p70S6K, PKCs (protein kinase C qθ, zζ, lλ), IKKβ (inhibitor of nuclear factor κB kinase β), and MAPK (*Gual et al., 2005*; *Taniguchi et al., 2006*). Consistent with our results, ROCK-mediated phosphorylation of IRS-1 at S632/635 was previously found to promote signaling throughput via IRS-1 and increased glucose uptake in adipocytes and mouse skeletal muscle (*Furukawa et al., 2005*; *Lee et al., 2009*). The uncoupling of IRS-1 S632/635 and S612 phosphorylation in MEFs and EWAT upon disruption of PDZ-RhoGEF may represent an additional means of achieving specificity in RhoA signaling. Intriguingly, PDZ-RhoGEF appears capable of directing ROCK activity towards IRS-1 S632/635 in fibroblasts and adipose tissue, whereas other GEFs may participate in modulating its activity towards S612 in other tissues. Alternatively, the distinct RhoGEFs may differentially activate ROCK family members (ROCK1/β and ROCKII/α), which in turn may harbor subtle, but important differences in specificity towards IRS-1.

## PDZ-RhoGEF as a modifier of insulin/IGF-1 signaling in control of energy expenditure and adipose tissue homeostasis

The lack of difference in the BAT mass, food consumption, UCP-1 expression, in BAT and IWAT, as well as the core body temperature in PDZ-RhoGEF KO mice, indicate that the adaptive thermogenesis is not dependent on PDZ-RhoGEF. Nevertheless, PDZ-RhoGEF deletion increased total oxygen consumption and total activity, largely due to an increase in the stereotype activity, but not the ambulatory activity. RER analysis indicates that while both wild type and PDZ-RhoGEF KO mice utilize carbohydrates and fat as fuel (0.8<RER<1.0), higher RER values in PDZ-RhoGEF KO mice suggest preferential use of carbohydrates as a fuel source in their skeletal muscles. Indeed, an increase in physical activity found in PDZ-RhoGEF KO mice may impact exercise- and contractility-mediated glucose transport in the skeletal muscle (*Richter and Hargreaves, 2013*) and in turn increase the use of carbohydrates as energy source. Remarkably, despite its reduced PI3K pathway activation in response to insulin, PDZ-RhoGEF KO EWAT displayed an increase in ECAR, revealing of the preferential use of glucose as an energy source in this tissue. This may in part be due to the compensatory, PI3K-independent mechanisms for glucose utilization in EWAT (*Chang et al., 2007*) and contribute to the maintenance of glucose homeostasis in PDZ-RhoGEF KO animals and their lean and metabolically healthy phenotype.

The phenotype of PDZ-RhoGEF-deficient mice partially resembles that of mice lacking other insulin signaling pathway components. For instance, IRS-1- or PKBα/Akt1-deficiency leads to decreased body fat (*Miki et al., 2001*; *Wan et al., 2012*). Similarly, adipose tissue-specific disruption of the insulin receptor (FIRKO) displayed reduced adiposity. Unlike the proliferative defect found in PDZ-RhoGEF KO adipose tissue development, the reduced adipose tissue size in FIRKO mice results from decreases in adipocyte size in the mutant adult fat pads (*Bluher et al., 2002*). Further confounding the comparison between the phenotypes arising from IR and PDZ-RhoGEF deletion is the contribution of PDZ-RhoGEF to signaling via the IGF-1 receptor (IGF-1R). IGF-1R is the dominant IR/IGF-1R family member expressed in the stromal-vascular cells of the developing adipose tissue, including progenitor cells and pre-adipocytes (*Entingh-Pearsall and Kahn, 2004*; *Nougues et al., 1993*; *Wright and Hausman, 1995*). Thus, in vivo, PDZ-RhoGEF impact on the proliferative capacity of adipogenic precursors likely relates to its function in signaling via the related IGF-1R, whereas in mature adipose tissue it may also affect IR signaling (*Figure 5A,B*). Notably, dual deletion of IGF-1 and insulin receptors in adipose tissues (FIGIRKO) results in reduced adiposity due to a decrease in adipocyte abundance (*Boucher et al., 2012*). Taken together, the correlation between full body deletion of PDZ-RhoGEF and deletion of IR/IGF-1R in adipose tissue further reinforces that modification of IR/IGF-1R signaling in adipose tissue plays a central role in controlling adipose tissue homeostasis. A mechanistic insight into the contribution of PDZ-RhoGEF-mediated control of energy expenditure and adipocyte numbers to reduced adiposity may require tissue-specific deletion of PDZ-RhoGEF in various mouse tissues.

### PDZ-RhoGEF, an obesity/T2D predisposition gene?

Deletion of PDZ-RhoGEF had no effect on food consumption when fed with either normal or high-fat diet. Under a normal dietary setting, PDZ-RhoGEF, via RhoA/ROCK activation, sensitized IRS1/PI3K signaling to insulin/IGF-1, thereby maintaining the proliferative potential of adipogenic progenitors and supporting adipose tissue expansion during development (*Figure 7E–I*). Of note, PDZ-RhoGEF may also act on mature adipocytes or other cell types to influence adipose tissue development and homeostasis. Adipocyte number is a key determinant of fat mass in humans, whereby individuals with higher number of adipocytes are prone to obesity (*Spalding et al., 2008*). Likewise, wild type mice, which displayed greater total adipocyte numbers, became obese and developed T2D when on prolonged HFD. Reduced adipose tissue expansion in PDZ-RhoGEF KO animals prevented these pathological consequences of the HFD. Of interest, under conditions of energy overload, PDZ-RhoGEF levels were elevated in the peripheral insulin-target tissues and accompanied by increased downstream signaling to p70S6K1 and enhanced negative feedback loop input into the insulin/insulin receptor signaling cascade(*Figure 7E–II*).

Our data firmly support the involvement of PDZ-RhoGEF in the genetics of fat mass regulation and predisposition to obesity and T2D, previously suggested by the association of the PDZ-RhoGEF SNP R1467H with the epidemiology of T2D (*Bottcher et al., 2008*; *Fu et al., 2007*; *Jin et al., 2010*; *Liu et al., 2011*; *Ma et al., 2007*). Further work on determining the nature of the PDZ-RhoGEF H1467R SNP variant, using functional reconstitution of PDZ-RhoGEF KO cells and mice, as well as biochemical characterization of its activity towards RhoA will be needed to fully understand its action in obesity and T2D predisposition.

## Materials and methods

### Animal work

All animal work was conducted according to the Policies and Guidelines of the Canadian Council on Animal Care and the Province of Ontario's Animals for Research Act.

### Analysis of body weight, fat tissue abundance and food intake

Wild type and PDZ-RhoGEF KO male mice were weaned at 4 weeks of age and placed on normal chow diet (NCD) (5% fat, TD LM-485, Harlan) until 17 weeks of age. Body weight was measured at 6-weeks of age. For the early-onset obesity group, mice were fed with high fat diet (HFD) (45% fat, TD-01435, Harlan, Table S1) beginning at 6 weeks of age for 16 weeks. Animals from the late-onset group were fed with HFD beginning at 18-weeks of age for 14 weeks. Body weights were recorded

weekly. White adipose tissues (WATs) masses were determined anatomically from various depots, epididymal (EWAT), retroperitoneal (RWAT), inguinal (IWAT), knee (KWAT), from HFD and NCD fed male mice. The final tissue weight was determined with or without normalization to the total body weight (n = 8–12). Food intake per mouse was determined over 15 days (n = 4–5).

## Magnetic resonance imaging (MRI) analysis

Magnetic resonance imaging of abdominal adiposity used a 7 T Biospec 70/30 USR (Bruker Corporation, Ettlingen, DE), equipped with a B-GA12 gradient coil insert and 7.2 cm inner diameter cylindrical quadrature RF volume coil. Mice were anaesthetized and maintained at 1.8% isoflurane, delivered through the bite bar of a dedicated slider bed (Bruker Corporation, Ettlingen, DE) with mice in prone orientation. A strong positive fat signal was generated using a respiratory-gated T2-weighted RARE pulse sequence, applied as a stack of 20–24 2D coronal images with full body coverage (echo time 36 ms, field-of-view 90 x 40 mm, matrix size 360 x 160, in-plane resolution 250 μm, slice thickness 1 mm). The pulse sequence repetition time and the total data acquisition time were determined by respiratory rate (scan time of approx. 6 min at respiratory rate of 30 breathes per minute). Fat pad volumes were assessed with medical image processing, analysis, and visualization (MIPAV) software (National Institute of Health (NIH) Center for Information Technology). Body fat indices were calculated by dividing adipose tissue volume by body weight.

## Indirect calorimetry

The maximal oxygen consumption rate ($VO_2$), heat production, respiratory exchange rate (RER $VO_2$/$VCO_2$), heat production (Kcal/hr), and activity in 9 to 14 month-old male WT and PDZ-RhoGEF KO mice were determined using the Oxymax system (Columbus Instruments). The method of indirect calorimetry scores counts as ambulation when the animal traverses the cage, breaking a series of IR beams in sequence. Repeated interruptions of the same IR beam do not incur ambulatory counts. All beam interruptions are scored as Total Activity (X TOT). Subtraction of ambulatory counts (X AMB) from the Total counts provides counts associated with stereotypy (grooming, scratching, etc.).

## Ex vivo oxygen consumption (OCR and ECAR)

Samples were prepared as described (*Dunham-Snary, et al., 2014*). In brief, Tissues were dissected from 8-week old mice and washed with Seahorse basal medium (DMEM) with 25 mM Glucose and 25 mM HEPES. Each tissue was sampled six times using a 2 mm UniCore Harris Punch (Whatman, Sigma-Aldrich), resulting in 2 mg tissue samples. The tissue punches (4 mg) were transferred to XF24 Islet Capture Microplate (Seahorse Bioscience, North Billerica, MA) and a screen was loaded on to each well. Each well was washed with running medium (DMEM/25 mM Glucose and without HEPES). OCR and ECAR were determined in running medium with Seahorse XF$^e$24 analyzer (Seahorse Bioscience).

## Ectopic expression of PDZ-RhoGEF in MEFs

PDZ-RhoGEF was introduced to MEFs using a retroviral packaging system. The viral particles containing myc-PDZ-RhoGEF was generated by Transfection of the Phoenix E retroviral packaging line and MEFs (passage 1) were infected and expanded for characterization.

## Metabolic studies

Fasting glucose tolerance test (GTT) and insulin tolerance test (ITT) were performed on age matched (8 month-old) wild type and PDZ-RhoGEF KO mice. For GTT, PDZ-RhoGEF wild type mice (n = 8) and KO mice (n = 8) were starved for 16 hr before intraperitoneal (IP) challenge with 1 g D-glucose/kg of body weight and their blood glucose monitored from the tail vein using an automated glucose monitor (Actusoft, Roche). ITT was performed following a three hour fast (n = 8). Human insulin (1 U/kg of body weight, Novo Nordisk) was given to the animals by intraperitoneal injection and the blood glucose level was measured every 15 min up to one hour as described above. Serum was collected from wild type (n = 6) and PDZ-RhoGEF KO mice (n = 8) after 16 hr of starvation for determination of fasting glucose, insulin, triglycerides, and glycerol. Glucose levels were measured by the automated glucose monitor (Actusoft, Roche). Insulin levels were determined by Ultrasensitive

insulin ELISA kit (Crystal Chem Inc., Downers Grove, IL) in triplicates. The levels of fasting triglycerides were determined by the serum triglyceride determination kit (Sigma-Aldrich, St. Louis, MO).

## Glucose uptake with isolated skeletal muscles

Whole extensor digitorum longus (EDL) and soleus muscles were isolated and incubated in the presence or absence of 100 nM insulin, and the uptake of 2-deoxyglucose (2-DG) was measured as described previously (*Rudich et al., 2003*)

## Adipocyte size distribution analysis

Epididymal white adipose tissues were prepared from NCD-fed 28-week old wild type and PDZ-Rho-GEF KO male mice and 42-week old mice from both genotypes with the least 24 weeks on HFD. Paraffin embedded tissues were sectioned in 4-μm-thick sections and stained with H&E. Adipocyte cell size was determined with Image J software (US National Institutes of Health). At least 100 cells were measured from each genotype (n = 4 for NCD; n = 6 for HFD).

## Isolation of mature adipocytes and stromal vasculature fraction

Mature adipocytes and stromal vascular fraction were isolated from epidydimal white adipose tissue as described (*Tang et al., 2008*). In brief, adipose tissues were harvested and minced in adipocyte isolation buffer (100 mM HEPES pH 7.4/120 mM NaCl/50 mM KCl/5 mM D-glucose/1 mM $CaCl_2$/ 1.5% fatty acid-free BSA) containing 0.8 mg/ml (125 U/ml) collagenase III (Worthington) at 37°C with shaking for 2 hr. Digestion was terminated by adding DMEM:F12 (1:1)/10% FCS/2 mM L-glutamine/ 1 mM sodium pyruvate/50 mM bβ-mercaptoethanol and digested fat tissues were passed through a 180-μm nylon mesh filter (Millipore). Stromal vascular fraction was collected using a 30 ml syringe with 14 G (80 mm long) needle to avoid the floating adipocytes, followed by centrifugation to collect stromal vascular fraction. Isolated stromal vascular fraction was subjected to FACS analysis to determine the adipocyte progenitor cell population. Adipocyte numbers were determined from approximately 100 mg of adipose tissues digested in 1 ml of adipocyte isolation buffer at 37°C with shaking at 125 rpm for 2 hr. After digestion, adipocytes in 10 μl suspension were counted using hematocytmeter. At least 200 adipocytes from each preparation were counted. Total number of adipocytes for each fad pad was calculated by converting total number per ml into total number per fad pad.

## Assays for mitotic clonal expansion and in vitro adipocyte differentiation

Post-confluent MEFs were subjected to the adipocyte differentiation induction medium (complete medium/1 mM dexamethasone/5 mg/ml insulin/0.5 mM isobutylmethylxanthine [IBMX]) or insulin (5 μg/ml) alone. Cell cycle re-entry and DNA synthesis was determined by [$^3$H]-thymidine incorporation at the indicated times. BrdU labeling was performed to monitor mitotic clonal expansion. MEFs were plated onto glass coverslips and grown to confluence. 15 hr after induction of differentiation, cells were pulse-labeled for 3 hr with BrdU (10 μM) and then transferred to BrdU-free induction medium. 72 hr later, cells were fixed and BrdU positive cells were detected by phycoerythrin-conjugated anti-BrdU antibody (Molecular Probe, Invitrogen) and counter stained with 4',6-diamidino-2-phenylindole (DAPI). For in vitro differentiation, post-confluent cells were maintained in induction medium for 2 days, and then were fed with complete medium supplement with 5 mg/ml of insulin every other day. At the end of the monitoring period (day 8), cells were fixed with 10% phosphate buffered formalin. Oil droplets were stained with Oil Red-O as described (*Tang et al., 2003*). Relative lipid content was determined by absorbance at 510 nm by spectrophotometer (Beckman/Coulter, DU800). Post-confluent EWAT ADSCs (from 16-week old male mice) were stimulated with insulin (5 ng/ml) and DNA synthesis was assessed by [$^3$H]-thymidine incorporation at each indicated time point.

## FACS analysis and isolation of white adipocyte progenitor cells from stromal-vascular fraction of EWATs

Isolated stromal-vascular fraction was resuspended in FACS media (2% FCS/PBS/0.01% $NaN_3$) and cells were stained with markers for adipose stem cell and preadipocytes (*Rodeheffer et al., 2008*). Antibody incubation was performed on ice for 20 min and fixed with 1% paraformaldehyde

overnight at 4°C. Samples were acquired on Canto II flow cytometry (BD Biosciences) and analyzed with FlowJo software (Tree Star, Inc., Ashland, OR). For all ex vivo analysis, PDZ-RhoGEF expression, progenitor cell proliferation, and PI3K signaling, adipogenic progenitor cells were sorted from adipogenic progenitor specific maker-labeled SVF suspensions with AriaII (Becton Dickson) cell sorter, and the purity of sorted cells was verified as >95% by rerunning the sorted population. Antibodies were purchased from eBioscience unless otherwise stated, including the following: biotin-lineage cocktail (), Sca1-Pacific Blue (Biolegend, San Diego, CA), Sca1-APC, CD34-Alexa-Fluro-700 (Biolegend), CD24-PE, CD31-PE-Cy7, Terr-119-APC-eFluro-780, biotin-CD45/streptavidin-PerCp, CD29-FITC (BD Biosciences), streptavidin-PE-Cy7, and anti-Dlk/Pref-1 (MBL).

## Progenitor cell culture

Sorted CD34+/CD29+/Sca1+ cell were expanded under hypoxic conditions (3% $O_2$, 5% $CO_2$) with MesenCult medium according to the manufacture's protocol (Stem cell Technology). The cells at passage 1–3 were used for differentiation, colony-forming assay (CFU-F), insulin stimulated BrdU uptake, and PI3K signaling experiments under normaxia condition in DMEM/10% FCS.

## In vivo BrdU labeling

In vivo BrdU labeling was done as described (*Staszkiewicz et al., 2009*). To quantify cell proliferation during early adipose tissue development, 6.5-day-old neonates were injected intraperitoneally with BrdU (Sigma-Aldrich) at a concentration of 50 µg/g of body weight, twice daily at 12 hr interval for 3.5 consecutive days. Animals were perfused through the left ventricle with physiological saline, and then 4% paraformaldehyde after a 3-week chase period and BrdU-labeled tissues were harvested for immunohistochemistry. BrdU-retaining cell population was quantified as percent of cells positive for BrdU (n = 6–9).

## Immunoblotting

Cells and tissues were disrupted in lysis buffer buffer (20 mM Tris (pH 7.5), 150 mM NaCl, 1 mM EDTA, 1 mM EGTA, 1% Triton X-100, 2.5 mM Na-pyrophosphate, 1 mM b-glycerophosphate, 1 mM $Na_3VO_4$, 1 mg/ml Leupeptin, 1 mM PMSF). Tissues were excised and lysed. Equal amounts of protein amounts were fractionated by NuPAGE (Novex) and analyzed using antibodies specific for phospho-Akt/PKBS473, PKB/Akt, and IR (Cell signaling, Danvers, MA), phospho-IRbY1150/1151and β-tubulin (Upstate). IRS-1 phosphorylation and protein-protein interaction analysis were performed by immunoprecipitating total IRS-1 with the anti-IRS-1 antibody (Cell Signaling) and immunoblotting with antibodies specific for phospho-IRS-1S632/635, phospho-IRS-1S612, IRS-1, p110α (Millipore). (Cell Signaling), phospho-tyrosine, p85α (Milliopore).

## RBD assay

RhoA activity was assessed by the RBD assay as described (*Diekmann and Hall, 1995*) using a monoclonal antibody against RhoA (Santa Cruz Biotechnology).

## Statistical analysis

For cell proliferation data, all results are presented as: mean + ± S. D of at least three independent experiments. For animal data, all results are presented as: mean ±_ S. E. M.. Statistical significance was calculated by a Student's t test using Excel software (*$p \leq 0.05$, **$p \leq 0.01$, ***$p \leq 0.001$).

# Supplementary methods

## Gene-targeting construct and generation of the PDZ-RhoGEF KO mice

The gene-targeting construct was generated using the FRT/loxP system. A replacement vector (pSPUC) was generated in which an FRT-flanked neo expression cassette was introduced into the intron between exon 1 and 2 of Arhgef11 (GeneBank accession # NM_001003912) and two loxP sites were engineered to flank the targeted exon 2, which created a null allele of PDZ-RhoGEF when Cre recombinase was introduced. The targeting construct was electroporated into embryonic stem cells (ES) from 129J/HSD strain. The homologous recombination and specific integration were confirmed by genomic Southern Blot. Subsequently, the homologous targeted ES clones were micro-injected

into blastocysts isolated from C57/BL6 strain and transferred into the 129J/HSD foster mothers. The germline transmission was confirmed by genomic Southern Blot. PDZ-RhoGEF heterozygous and null animals were generated by crossing with a Cre transgenic line, B6.C-Tg(CMV-cre)1Cgn/J (Jackson laboratory) that expresses Cre recombinase globally under the transcriptional control of a human cytomegalovirus (CMV) promoter. The deletion of exon 2 was confirmed by genomic PCR with specific primers, Southern Blot with flanking probe, and Western Blot with anti-PDZ-RhoGEF antibody. The congenic C57BL6 PDZ-RhoGEF animals were generated by backcrossing C57/BL6:129J/HSD six to ten generation with C57/BL6 strain.

## Generation of antibody against PDZ-RhoGEF

The N-terminal fragment of human PDZ-RhoGEF (aa1-735, PDZ-RhoGEFΔC) was subcloned into a bacterial expression vector pGEX-4T.1 (Amersham Pharmacia Biotech) to generate a recombinant fusion protein GST (Glutathione-S-Transferase)-PDZ-RhoGEFΔC. Purified recombinant fusion protein was used to make a polyclonal rabbit anti-PDZ-RhoGEF antibody (Department of Comparative Medicine, University of Toronto). The anti-PDZ-RhoGEF antibody was further purified with GST-sepharose affinity gel filtration.

## Generation of mouse embryonic fibroblasts (MEFs)

MEFs were derived from congenic C57BL6 mouse embryos at day 14.5 (E14.5). All the experiments were carried out using early passage of primary MEFs (passage 3) derived from male PDZ-RhoGEF KO embryos and the littermate PDZ-RhoGEF$^{+/+}$ embryos (WT) as control. Genotype and gender were determined by genomic PCR.

## Cell proliferation

For assessment of mitogen-stimulated cell proliferation, cells were plated and starved of serum for 48 hr and proliferation scored by measuring [3H]-thymidine uptake in response to mitogens, IGF-1 (80 ng/ml), LPA (20 mM) or FCS (10%). Post-confluent ADSCs were stimulated with insulin (5 ng/ml) and DNA synthesis was assessed by [3H]-thymidine incorporation at each indicated time point. Cell proliferation was also determined by cell counting (Coulter counter, Beckman Coutler).

## Cell migration and actin cytoskeleton assays

Cell migration was evaluated using both wound healing and transwell migration assays. Briefly, for the wound healing assay, MEFs were plated and grown on a 24-well plate coated with 0.1 mg/ml of poly-D-lysine to form a confluent monolayer and serum starved for 48 hr. Wound was generated by scraping the monolayer with a pipette tip in the presence 10 mM of mitomycin C and migration was induced by adding serum-free medium, FCS (10%), IGF-1 (100 ng/ml), LPA (20 µM) in the presence with mitocycin C. Cells were monitored for a 24 hr period, then fixed in 4% paraformaldehyde and stained with 0.1% crystal violet. For transwell assay, MEFs in a serum-free medium were plated onto the upper chamber of transwell plares (Corning, New York), and IGF-1 (100 ng/ml), and was added to the lower chamber. After 6 hr, the cells from the inside of chamber were removed with cotton swab and the cells on the bottom of the filter were fixed with 4% paraformaldehyde and cells stained with 2 µg/ml DAPI in 0.1% Triton X-100/PBS. To assess stress fiber formation, MEFs were plated on to coverslips in 24-well plates and serum starved for 48 hr. Cells were fixed with 4% paraformaldehyde 5 minutes after addition of FCS (10%), IGF-1 (100 ng/ml) and LPA (20 µM). Fixed cells were permeabilized in 0.1% Triton X-100/PBS, and stress fiber was stained with Rhodamine-Phalloidin (1 U/ml) in 0.1% Triton X-100 PBS. Stained cells were analyzed by confocal microscopy (Zeiss LSM510).

## Real-time quantitative RT-PCR

Total RNA was extracted from frozen tissue samples using Trizol (Invitrogen), followed by RNeasy kit (Qiagen) to further purify RNA. Complementary DNA was synthesized from total RNA with the SuperScript transcriptase III and random hexamer primers (Invitrogen, Carlsbad, CA). The real-time polymerase chain reaction (PCR) measurement of individual cDNAs was performed using SYBR green dye to measure duplex DNA formation with ABI 7500 Real-Time PCR System (Applied Biosciences, Waltham, MA). The expression was normalized to the expression of either 18S

ribosomal, glyceraldehyde-3-phosphate dehydrogenase (GAPDH) RNA, or mouse TATA box binding protein 1 (mTBP1). The primers used in the real time RT-PCR were listed as following:

Pref-1
forward: 5′-GACCCACCCTGTGACCCC-3′
reverse: 5′-CAGGCAGCTCGTGCACCCC-3′
UCP-1
forward: 5′-AGGCTTCCAGTACCATTAGGT-3′
reverse: 5′-CTGAGTGAGGCAAAGCTGATT-3′
18S
forward: 5′AAACGGCTACCACATCCAAG3′
reverse: 5′CCTCCAATGGATCCTCGTTA3′
GAPDH
forward: 5′-AACTTTGGCATTGTGGAAGG-3′
reverse: 5′-ACACATTGGGGGTAGGAACA-3′
G-6-Pase
forward: 5′-GAAGGCCAAGAGATGGTGTGA-3′
reverse: 5′-TGCAGCTCTTGCGGTACATG-3′
DGAT-1
forward: 5′-GGCCTGCCCCATGCGTGATTAT-3′
reverse:  5′-CCCCACTGACCTTCTTCCCTGTAGA-3′
GLUT2
forward: 5′-GGCTAATTTCAGGACTGGTT-3′
reverse:  5′-TTTCTTTGCCCTGACTTCCT-3′
ChREBP
forward: 5′-CTG GAC CGA CCC CTC TCC TT-3′
reverse: 5′-TTG TGG GTG CAG GAA GCG TA-3′
mTBP1
forward: 5′-GGCCTCTCAGAAGCATCACTA-3′
reverse: 5′-GCCAAGCCCTGAGCATAA-3′

## Quantification of adipose tissue macrophages

To quantify macrophages at early adipose tissue development, BrdU-labeled tissue sections were with an anti-F4/80 monoclonal antibody. For each section from each individual mouse, three different high-power fields from individual mouse were analyzed. The total number of adipocytes and the number of nuclei of F4/80-expressing cells were counted for each field using Image J (NIH). The fraction of F4/80-expressing cells for each sample was calculated as the sum of the number of nuclei of F4/80-expressing cells divided by the total number of adipocytes in each section of a sample.

## Histology of white adipose tissues and liver

Dissected livers and EWATs were fixed in formalin, embedded in paraffin, sectioned in 4 µm slices, and stained with hematoxylin-eosin (H&E).

## PDZ-RhoGEF expression in various tissues

PDZ-RhoGEF expression was determined by immunoblotting. Protein lysates were prepared from various tissues from wild type male mice, subjected to 3~8% Tris-Acetate NuPAGE gel (Invitrogen) and immunoblotted with anti-PDZ-RhoGEF and anti-actin antibodies.

## Assays for mitotic clonal expansion, and in vitro adipocyte differentiation

Post-confluent MEFs were subjected to the adipocyte differentiation induction medium (complete medium/1 mM dexamethasone/5 mg/ml insulin/0.5 mM isobutylmethylxanthine (IBMX)) or insulin (5 mg/ml) alone. Cell cycle re-entry and DNA synthesis was determined by [3H]-thymidine incorporation at the indicated times. BrdU labeling was performed to monitor mitotic clonal expansion. MEFs were plated onto glass converslips and grown to confluence. 15 hr after induction of differentiation, cells were pulse-labeled for 3 hr with BrdU (10 mM) and then transferred to BrdU-free

induction medium. 72-hr later, cells were fixed and BrdU positive cells were detected by phycoery-thrin-conjugated anti-BrdU antibody (Molecular Probe, Invitrogen) and counter stained with 4',6-dia-midino-2-phenylindole (DAPI). For in vitro differentiation, post-confluent cells were maintained in induction medium for 2 days, and then were fed with complete medium supplement with 5 mg/ml of insulin every other day. At the end of the monitoring period (day 8), cells were fixed with 10% phosphate buffered formalin. Oil droplets were stained with Oil Red-O as described (*Tang et al., 2003*). Relative lipid content was determined by absorbance at 510 nm by spectrophotometer (Beck-man/Coulter, DU800). Post-confluent ADSCs were stimulated with insulin (5 ng/ml) and DNA synthesis was assessed by [3H]-thymidine incorporation at each indicated time point. The number of mature adipocytes derived from MEFs was averaged of three independent fields after Oil Red-O staining (n = 3-–4).

## Acknowledgements

We wish to thank A Wakeham and A You-Ten for their help with embryonic stem cell injections, as well as Dr. R Dowling and Dr. C DeLuca for critical reading of the manuscript. This work was supported by the grants from the Canadian Cancer Society to TM (grant # 20003), Canadian Institutes for Health Research to AK (grant # 12601) and Canadian Breast Cancer Foundation (Ontario chapter) to VSTM and AK each hold a Canada Research Chair.

## Additional information

### Funding

| Funder | Grant reference number | Author |
| --- | --- | --- |
| Canadian Breast Cancer Foundation | Operating grant | Vuk Stambolic |
| Canadian Cancer Society Research Institute | 2011700891 | Vuk Stambolic |

The funders had no role in study design, data collection and interpretation, or the decision to submit the work for publication.

### Author contributions

YJC, Conception and design, Acquisition of data, Analysis and interpretation of data, Drafting or revising the article; SP, PD, Conception and design, Drafting or revising the article; SM, Acquisition of data, Analysis and interpretation of data, Drafting or revising the article; WF, NL, AK, Acquisition of data, Analysis and interpretation of data; LZ, Acquisition of data, Drafting or revising the article; MW, Analysis and interpretation of data; ZH, Acquisition of data; PJB, Analysis and interpretation of data, Drafting or revising the article; TM, Conception and design; VS, Conception and design, Analysis and interpretation of data, Drafting or revising the article

### Ethics

Animal experimentation: All animal work was conducted according to the Policies and Guidelines of the Canadian Council on Animal Care and the Province of Ontario's Animals for Research Act. The protocol was approved by the Animal Care Committee of Princess Margaret Cancer Center at University Health Network (permit Number:933 and 2176).

## Additional files

### Supplementary files

• Supplementary file 1. Composition of the high-fat diet.

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
