## [Decision Letter]

Thank you for sending your work entitled "PDZ-RhoGEF Governs Susceptibility to Diet-induced Obesity and Type II Diabetes" for consideration at *eLife*. Your article has been favorably evaluated by Fiona Watt (Senior Editor), Amy Wagers (Reviewing Editor), and three reviewers.

The Reviewing Editor and the other reviewers discussed their comments before we reached this decision, and the Reviewing editor has assembled the following comments to help you prepare a revised submission.

While all reviewers recognized the importance of these findings with regard to the role of PDZ-RhoGEF in white adipose biology and diet-induced obesity, specific concerns were raised, with which all reviewers agreed, that the manuscript in its current form does not sufficiently explain the energy expenditure and insulin signaling phenotypes in the PDZ-RhoGEF KO mice. We believe this issue can be addressed in a reasonable time frame through the provision of additional data to clarify which tissues contribute to energy expenditure phenotype and by which mechanism this phenotype occurs. Please see specific suggestions below:

1) Please include additional signaling analysis in different tissues and specific cell types (particularly in adipocyte precursor cells, though analysis in liver would also be helpful) to solidify the link between your in vitro and in vivo studies. The current use of extremely heterogeneous populations of MEFs or ADSCs to study adipogenesis in vitro (rather than analysis of adipocyte precursor cells as in Figure 3) makes it difficult to connect the mechanisms suggested by the in vitro studies to the in vivo loss of adipose tissue.

2) Similarly, the authors should analyze BrdU incorporation, IGF signaling and PDZ-RhoGEF expression in specific cell subsets that are relevant to the in vivo physiology of the PDZ-RhoGEF KO mice (i.e. adipocyte precursor cells), rather than in MEFs or adipose tissue generally, to provide greater mechanistic insights.

3) Please include additional data on energy expenditure: The implications of the higher RER is not demonstrated or discussed. The KO mice appear to be using significantly more glucose, although glucose uptake by skeletal muscle is similar. Are the white fat depots utilizing more glucose in these mice? What is the oxygen consumption rate in the fat pads? What about glucose use in the liver? It is also not convincing that the mice have any difference in energy expenditure, as the mice have little increase in oxygen consumption (only one time point is statistically significant) and no significant change in locomotor activity (despite some small trends with high p-values).

4) Similarly, with HFD, there are changes in EDL glucose uptake and lipid accumulation in the liver, making it unclear what contribution the changes in metabolism in these tissues have to phenotype of the mice. More metabolic analyses should be performed in these tissues, as the reduced adiposity may not be the major factor accounting for improved insulin sensitivity. Although fat accumulation in the liver may be a consequence of the changes in the fat, this may not necessarily be the case. It is hard to make any conclusions because there is not much studied in the livers regarding their metabolism. It is also unclear how glucose uptake is being altered in the EDL since PDZ-RhoGEF is not detected in this tissue (unless the levels are increased after HFD?) – this suggests there may be crosstalk occurring, which should be acknowledged in the manuscript.

5) Several conclusions appear to be drawn based on data that is not statistically significant (Figure 3—figure supplement 4) or that lacks sufficient sample size (sample size of only 1 or 2 is insufficient). Please add additional data or amend conclusions.

[Editors' note: further revisions were requested prior to acceptance, as described below.]

Thank you for resubmitting your work entitled "The Rho-guanine nucleotide exchange factor PDZ-RhoGEF Governs Susceptibility to Diet-induced Obesity and Type 2 Diabetes" for further consideration at *eLife*. Your revised article has been favorably evaluated by Fiona Watt (Senior Editor), a Reviewing Editor, and three reviewers. The manuscript has been substantially improved but there is one remaining issue that needs to be addressed before acceptance, as outlined below:

The authors have added new experiments to address the role of adipocyte progenitor cell proliferation in the phenotype of the PDZ-RhoGEF KO mice; however, the most specific assays (new Figure 3) are limited to effects in culture, and it remains possible that the in vivo phenotype could relate to effects in mature cells as well. Please edit the text to tone down the conclusion regarding the role of adipocyte precursor cell proliferation, and add to the Discussion explicit consideration that PDZ-RhoGEF may act on multiple cell types including precursor cells, mature adipocytes or other cell types to influence adipose tissue development and homeostasis. No new experiments are requested – this is merely a text edit to provide clarity for readers that a more detailed study of adipose precursor cells in this context is warranted.

---

## [Author Response]

*While all reviewers recognized the importance of these findings with regard to the role of PDZ-RhoGEF in white adipose biology and diet-induced obesity, specific concerns were raised, with which all reviewers agreed, that the manuscript in its current form does not sufficiently explain the energy expenditure and insulin signaling phenotypes in the PDZ-RhoGEF KO mice. We believe this issue can be addressed in a reasonable time frame through the provision of additional data to clarify which tissues contribute to energy expenditure phenotype and by which mechanism this phenotype occurs. Please see specific suggestions below: 1) Please include additional signaling analysis in different tissues and specific cell types (particularly in adipocyte precursor cells, though analysis in liver would also be helpful) to solidify the link between your in vitro and in vivo studies. The current use of extremely heterogeneous populations of MEFs or ADSCs to study adipogenesis in vitro (rather than analysis of adipocyte precursor cells as in Figure 3) makes it difficult to connect the mechanisms suggested by the in vitro studies to the in vivo loss of adipose tissue.*

We have directly addressed this request by analyzing signaling in adipocyte precursors and peripheral insulin-target tissues. Our new data indicate that insulin/IGF-1 signaling is impaired in PDZ-RhoGEF-deficient cultured FACS- isolated cell population (Lin^–^/CD29^+^/CD34^+^/Sca1^+^), which represents the pool of adipogenic progenitor cells (new Figure 4). We have also evaluated the impact of PDZ-RhoGEF on insulin/IGF-1 signaling in the liver and the skeletal muscle and found that its’ deletion produced no effects (Figure 5 moved forward from the Figure 5—figure supplement 1). These results further support the assertion that PDZ-RhoGEF impacts insulin/IGF-1 signaling specifically in white adipose tissue. The revised text was amended to reflect the new data.

*2) Similarly, the authors should analyze BrdU incorporation, IGF signaling and PDZ-RhoGEF expression in specific cell subsets that are relevant to the in vivo physiology of the PDZ-RhoGEF KO mice (i.e. adipocyte precursor cells), rather than in MEFs or adipose tissue generally, to provide greater mechanistic insights.*

To assess this, we further examined the effect of PDZ-RhoGEF loss on the progenitor cell proliferation. Similar to MEFs and ADSCs, deletion of PDZ- RhoGEF led to reduced proliferative capacity of adipogenic progenitor cells (CD34^+^CD29^+^Sca1^+^), as demonstrated by measuring BrdU incorporation (new Figure 3). The revised text was amended to reflect the new data.

*3) Please include additional data on energy expenditure: The implications of the higher RER is not demonstrated or discussed. The KO mice appear to be using significantly more glucose, although glucose uptake by skeletal muscle is similar. Are the white fat depots utilizing more glucose in these mice? What is the oxygen consumption rate in the fat pads? What about glucose use in the liver? It is also not convincing that the mice have any difference in energy expenditure, as the mice have little increase in oxygen consumption (only one time point is statistically significant) and no significant change in locomotor activity (despite some small trends with high p-values).*

In the revised manuscript, we have considerably extended the analysis of energy expenditure. Compared to wild type, the oxygen consumption and the respiratory exchange rates (RER) in PDZ-RhoGEF^-/-^ mice were increased at time points towards the end of the light cycle (revised Figure 2, Figure 2—figure supplement 2). Consistently, the isolated EWATs from both genotypes displayed similar basal OCR, however, PDZ-RhoGEF-deficient EWATs reached a higher maximal OCR than the wild types upon the injection of carbonyl cyanide-4-(trifluoromethoxy)-phenylhydrazone (FCCP), indicating greater maximal respiratory capacity (new Figure 2). Further, PDZ-RhoGEF^-/-^ EWATs displayed a higher extracellular acidification rate (ECAR), suggesting their preference for using glucose as a primary energy source (new Figure 2).

In the liver, glucose transport is glucose level-dependent. As a surrogate measure of glucose transport capacity in the liver, we examined the glucose-dependent gene expression of glucose transporter 2 (Glut2) and carbohydrate response binding protein (ChREBP), two molecules involved in glucose transport and glucose-dependent induction of glycolytic and lipogenic enzyme gene

expression, respectively. Neither ChREBP nor Glut2 gene expression were affected in the PDZ-RhoGEF^-/-^ livers (new Figure 6—figure supplement 1), suggesting that PDZ-RhoGEF had no effect on glucose utilization in liver.

In the muscle, we have previously shown that the deletion of PDZ-RhoGEF had no effect on insulin-dependent glucose transport in the skeletal muscle under normal conditions (Figure 6—figure supplement 1). Skeletal muscles regulate glucose transport through both insulin-dependent and insulin-independent mechanisms (exercise/skeletal muscle contractility) (Richter and Hargreaves, 2013). An increase in physical activity found in PDZ-RhoGEF^-/-^ mice (new Figure 2) may impact contractility-mediated glucose transport and in turn increase the use of carbohydrates as energy source.

Taken together, this new data further support the notion that increased RER is a consequence of preferential use of glucose as an energy source in PDZ- RhoGEF^-/-^ mice. The revised text was amended to reflect the new data.

*4) Similarly, with HFD, there are changes in EDL glucose uptake and lipid accumulation in the liver, making it unclear what contribution the changes in metabolism in these tissues have to phenotype of the mice. More metabolic analyses should be performed in these tissues, as the reduced adiposity may not be the major factor accounting for improved insulin sensitivity. Although fat accumulation in the liver may be a consequence of the changes in the fat, this may not necessarily be the case. It is hard to make any conclusions because there is not much studied in the livers regarding their metabolism. It is also unclear how glucose uptake is being altered in the EDL since PDZ-RhoGEF is not detected in this tissue (unless the levels are increased after HFD?) – this suggests there may be crosstalk occurring, which should be acknowledged in the manuscript.*

We performed further glucose and lipid metabolism in the liver analyses on our mice on HFD. In wild type livers on HFD, but not those from PDZ-RhoGEF^-/-^ mice on the same diet, mRNA levels of diglyceride acyltransferase 1 (DGAT1), an enzyme that participates in triglyceride synthesis were increased (new Figure 6) as were the mRNA levels of glucose-6-phosphatase (G-6-Pase), a key enzyme involved in the final step of gluconeogenesis and glycogenolysis in the liver (new Figure 6), suggesting that increased glucose level and lipid accumulation in wild type mice on HFD may be partly due to elevated hepatic glucose production (Table 1).

Regarding the EDL, we expanded our analysis of PDZ-RhoGEF in wild type mice on HFD. Under normal conditions, compared to EWAT and liver, the levels of PDZ-RhoGEF are considerably lower in white skeletal muscle, such as the EDL. Nevertheless, similar to EWAT, PDZ-RhoGEF protein levels in wild type animals were elevated in the liver and the EDL after prolonged HFD, accompanied by increased T389 phosphorylation of p70S6K (new Figure 7) and insulin resistance (Figure 7).

The revised text was amended to reflect the new data.

*5) Several conclusions appear to be drawn based on data that is not statistically significant (Figure 3—figure supplement 4) or that lacks sufficient sample size (sample size of only 1 or 2 is insufficient). Please add additional data or amend conclusions.*

The original analysis displayed absolute numbers from a representative experiment. We have now recalculated the data using relative measures of proliferation upon PDZ-RhoGEF expression (necessitated by variable thymidine uptake from experiment to experiment) from three independent experiments (revised Figure 3—figure supplement 5). Moreover, we have also explored 3T3L1 proliferation when PDZ-RhoGEF is knocked down. Judging by total cell counts and BrdU incorporation, decrease in PDZ-RhoGEF protein levels impaired both full media and insulin-stimulated proliferative capacity of 3T3-L1 (new Figure 3—figure supplement 4, corroborating our previous results.

[Editors' note: further revisions were requested prior to acceptance, as described below.]

The authors have added new experiments to address the role of adipocyte progenitor cell proliferation in the phenotype of the PDZ-RhoGEF KO mice; however, the most specific assays (new Figure 3) are limited to effects in culture, and it remains possible that the in vivo phenotype could relate to effects in mature cells as well. Please edit the text to tone down the conclusion regarding the role of adipocyte precursor cell proliferation, and add to the Discussion explicit consideration that PDZ-RhoGEF may act on multiple cell types including precursor cells, mature adipocytes or other cell types to influence adipose tissue development and homeostasis. No new experiments are requested – this is merely a text edit to provide clarity for readers that a more detailed study of adipose precursor cells in this context is warranted.

In response to the request to tone down the assertions regarding the impact of PDZRhoGEF on adipocyte progenitors only, we have deleted the second half of the sentence in the subsection “Reduction of adipocyte and adipocyte progenitor abundance in PDZ- RhoGEF KO EWATs” – “PDZ-RhoGEF KO EWAT displayed fewer BrdUretaining cells than the wild type EWAT, indicative of impaired proliferation during early adipose tissue development (Figure 3)” – to read “PDZ-RhoGEF KO EWAT displayed fewer BrdU-retaining cells than the wild type EWAT, indicative of impaired proliferation (Figure 3)”. The text of the Discussion was also amended to include this sentence: “Of note, PDZ-RhoGEF may also act on mature adipocytes or other cell types to influence adipose tissue development and homeostasis”. As there are no other sections in which we relate PDZRhoGEF to the adipose precursors, we trust that this satisfies the request by the reviewers.